



# Thirty Years of Arctic Primary Marine Organic Aerosols: Patterns, Seasonal Dynamics, and Trends (1990–2019)

Anisbel Leon-Marcos[1], Manuela van Pinxteren[2], Sebastian Zeppenfeld[2], Moritz Zeising[3],

Astrid Bracher[3,4], Laurent Oziel[3], Ina Tegen[1], and Bernd Heinold[1]

[1]Modeling of Atmospheric Processes Department, Leibniz Institute for Tropospheric Research, 04318 Leipzig, Germany

[2]Atmospheric Chemistry Department (ACD), Leibniz Institute for Tropospheric Research, 04318 Leipzig, Germany

[3]Alfred Wegener Institute, Helmholtz Center for Polar and Marine Research, Bremerhaven, Germany

[4]Institute of Environmental Physics, University of Bremen, Bremen, Germany

**Correspondence:** Bernd Heinold (heinold@tropos.de)

**Abstract.** Changing Arctic climate patterns have led to sea ice retreat, impacting ocean and atmospheric dynamics as well as marine ecosystems. Reduced sea ice cover likely enhances emissions of primary marine aerosols (sea salt and organic matter) via bubble bursting, potentially amplifying aerosol-cloud interactions. Moreover, primary marine organic aerosol (PMOA) production is closely linked to variations in marine biological productivity. This study examines the emission patterns, seasonality,

and historical trends of key biomolecule groups (dissolved carboxylic acid-containing polysaccharides (PCHO), dissolved combined amino acids (DCAA), and polar lipids (PL)) within the Arctic Circle from 1990 to 2019. Surface ocean concentrations of these groups are derived from a biogeochemistry model and used as input to the aerosol-climate model ECHAM-HAM. Results indicate that the strong seasonality in biomolecule concentrations and PMOA emissions is driven by marine productivity and sea salt emissions. These quantities peak from May to September, coinciding with the phytoplankton bloom and seasonal

sea ice minimum. Accumulated aerosol emissions and burdens over the Arctic increased by at least 7 % and 4 %, respectively, between the first and second halves of the study period. Summer trend analysis (June–August) reveals a strong reduction in sea ice that correlates with rising concentrations of organic groups in seawater in the inner Arctic. Positive emission anomalies have become more frequent over the past 15 years, indicating an overall upward trend. Average PMOA production has increased by 0.8 % per year since 1990. However, changes vary across biomolecular types and Arctic subregions, with PCHO showing the

largest relative increase.



## 1 Introduction

The Arctic region is undergoing drastic changes as surface air temperatures are increasing more rapidly than those for the rest of the world (Wendisch et al., 2019; Rantanen et al., 2022; Wendisch et al., 2023). This phenomenon, known as Arctic amplification, is driven by several feedback mechanisms (Block et al., 2020; Wendisch et al., 2023). One key process is the

sea ice–albedo feedback, in which the decline of highly reflective sea ice and snow surfaces contributes to further warming and melting sea ice (Serreze and Barry, 2011). Particularly, the unprecedented decline in sea ice area over the past 30 years presents an urgent call for research (Johannessen et al., 2004). Since the positive ice-albedo feedback mechanism in the Arctic has contributed to warming the ocean, the open water season has consequently extended (Perovich et al., 2007; Stammerjohn et al., 2012; Arrigo and van Dijken, 2015). The retreating sea ice also impacts the marine biological activity by a complex

chain of processes linked to light availability, fresh nutrient supply and vertical mixing (Ardyna and Arrigo, 2020; Nöthig et al., 2020). As a result, the distribution and magnitude of phytoplankton blooms, as well as the duration of the growing season, have notably changed in the last decades (Arrigo et al., 2008; Arrigo and van Dijken, 2011; Ardyna and Arrigo, 2020). These factors modify the total primary production and determine regional differences within the Arctic (Arrigo et al., 2008; Kahru et al., 2011; Aksenov et al., 2011; Fernández-Méndez et al., 2015; Cherkasheva et al., 2025).

This likely also affects the Arctic aerosol burden, which has a significant contribution from local marine sources (Moschos et al., 2022). Here, sea spray aerosol, primarily generated through bubble bursting of breaking waves driven by wind action on the sea surface, is a major contributor during the Arctic summer (Leck et al., 2002; Deshpande and Kamra, 2014; Heintzenberg et al., 2015; Willis et al., 2017; Lawler et al., 2021; Gu et al., 2023). Organic surfactants present in seawater attach to rising bubbles and are released into the atmosphere together with sea salt (Facchini et al., 2008; Keene et al., 2007; Gantt et al., 2011;

Gantt and Meskhidze, 2013). The organic particles originated through this mechanism are known as primary marine organic aerosol (PMOA) (Facchini et al., 2008; Gantt et al., 2011; de Leeuw et al., 2014). As a result of the changing climate conditions, the melting sea ice leads to new, extensive areas of open water and ice fractures, where wind-driven sea spray emissions could occur. Additionally, the relationship between PMOA production and the release of ocean surface organic components through biological processes, suggests that variations in marine productivity could affect the marine aerosol emissions. This,

in turn, potentially has far-reaching consequences for aerosol-cloud interactions and associated climate effects in the Arctic. Observations have widely documented the important role of local marine sources (Russell et al., 2010; Frossard et al., 2014; May et al., 2016; Kirpes et al., 2019; Lawler et al., 2021; Zeppenfeld et al., 2019, 2023; Rocchi et al., 2024) and the relevance of PMOA for cloud formation in the Arctic. The presence of marine organics in aerosol has been linked to marine biological activity as a correlation with phytoplankton proxies (chlorophyll-$a$) and measured organic compounds in seawater (Leck and

Bigg, 2005a; O'Dowd et al., 2008; Russell et al., 2010; Rinaldi et al., 2013; May et al., 2016; Zeppenfeld et al., 2023). In addition, the capability of PMOA to act as cloud condensation nuclei (CCN), has been explained by the strong dependence found between CCN population and insoluble organic aerosols linked to the composition of the marine surface microlayer (the top-most ocean layer at the ocean-atmosphere interface) (Leck and Bigg, 2005a). Moreover, repeated evidence of biological ice nucleating particles (INP) in relation to local marine emissions in the Arctic and at Nordic Seas stations has been extensively





reported (Wilson et al., 2015; Irish et al., 2017; Creamean et al., 2019; Wilbourn et al., 2020; Hartmann et al., 2021; Creamean et al., 2022; Porter et al., 2022; Sze et al., 2023).

The representation of PMOA emissions in aerosol-climate models considers the same principles found in observations. Available emission parametrizations for estimating the organic mass fraction in sea spray, follow either a chl-$a$ based empirical formulation (O'Dowd et al., 2008; Gantt et al., 2011; Rinaldi et al., 2013) or an organic-class-resolved approach that accounts

for the physico-chemical characteristics of ocean biomolecules (Burrows et al., 2014). Both types of schemes have been implemented and evaluated in global models (Gantt and Meskhidze, 2013; Huang et al., 2018; Zhao et al., 2021; Leon-Marcos et al., 2024). Nonetheless, the analysis of the PMOA as species-resolved organic groups (e.g, polysaccharides, proteins, and lipids) could provide additional evidence of potential differences in marine organic aerosol abundance. Recent findings, in Arctic measurements, confirm the high enrichments of carbohydrates in aerosols, which were also detected in the surface microlayer

of the marginal ice zone and aged melt ponds (Zeppenfeld et al., 2023). Similarly, a notable contribution of glucose, which could be considered as a proxy for ice nucleating activity (Zeppenfeld et al., 2019), has been measured in sea spray aerosol north of 80°N (Rocchi et al., 2024). Therefore, the critical role of PMOA emissions, transport patterns and evolution under the rapid changing climate should be thoroughly studied separately for individual species.

The effect of retreating Arctic sea ice on sea spray emissions has been discussed to some extent, and model results point to

an increase in sea salt aerosol concentration in the following decades (Struthers et al., 2011; Gilgen et al., 2018; Lapere et al., 2023). In light of the increasing fraction of sea ice cracks, leads, melt ponds and the marginal ice zone (Rolph et al., 2020; Zhang et al., 2018; Willmes and Heinemann, 2015), they are currently considered a relevant source of local emissions via bubble bursting (May et al., 2016; Kirpes et al., 2019; Lapere et al., 2024). Insights on the organic contribution from these marine source have been provided in recent studies (Kirpes et al., 2019; Zeppenfeld et al., 2023; Wang et al., 2024). Nevertheless, the

spatio-temporal distribution among organic compounds in water bodies is not uniform and strongly depends on the marine biological origin of the considered biomolecule groups in seawater (Burrows et al., 2014; Leon-Marcos et al., 2024). Furthermore, the interplay between marine sources and the loss of sea ice as well as their relevance for PMOA and mixed-phase clouds and, thus, for the climate in the Arctic remains unclear (Wendisch et al., 2023). To a large extent, this is due to remaining uncertainties and limitations in the understanding and representation of the life cycle and aerosol-cloud effects of PMOA in

aerosol-climate and Earth System Models (ESM) (Taylor et al., 2022). Based on observational evidence of marine biogenic INP particles predominance, their consideration in ESM will potentially improve the model representation of clouds (Schmale et al., 2021).

Given the biomolecule physico-chemical characteristics, some groups are selectively aerosolized (lipids) whereas others have higher INP potential (polysaccharides and proteins) (Facchini et al., 2008; Burrows et al., 2014; Alpert et al., 2022). Such

disparities are pronounced in the Arctic by the complex dynamical changes of sea ice and atmospheric conditions. Hence, the response of PMOA species abundance and indirect climate impact presumably respond differently to changes in the fragile marine ecosystem. Understanding how marine biomolecules and their organic contributions to aerosols have evolved under the changing Arctic climate is therefore essential. To our knowledge, however, a species-resolved trend analysis of marine organic groups in seawater and aerosols has not been performed.



In this study, we aim to unravel how the interplay of emission drivers have determined the evolution of PMOA species within the Arctic circle (66 °N-90 °N) from 1990 to 2019. For the simulation experiments, we use the model configuration as described in Leon-Marcos et al. (2024) for the aerosol-climate model ECHAM6.3-HAM2.3 (Tegen et al., 2019). As relevant for the PMOA emissions, the following highly abundant biomolecule groups in seawater are taken into account (dissolved carboxylic acidic containing polysaccharides (PCHO), dissolved combined amino acids (DCAA), and polar lipids (PL)). The

OCEANFILMS (Organic Compounds from Ecosystems to Aerosols: Natural Films and Interfaces via Langmuir Molecular Surfactants, Burrows et al. (2014)) scheme, recently implemented into the ECHAM-HAM model, allows for accounting for the organic fraction of these groups in nascent sea spray and simulating the aerosol transport, transformation, and removal processes.

## 2 Methods


This study examines the patterns, seasonal dynamics, and trends of primary marine organic aerosols (PMOA) in the Arctic region using results from a comprehensive marine biogeochemical model that simulates key oceanic biomolecules and their associated production and sink processes. These results are used in simulations of a global aerosol-climate model to represent emissions and transport of PMOA, focusing specifically on key species groups. The detailed technical description of the asso-

ciated model development, configuration, and input data is provided by Leon-Marcos et al. (2024). All abbreviations used in the present study referring to marine groups and aerosol components are in accordance to the definitions by Leon-Marcos et al. (2024) and are listed in Table A1. This analysis spans a 30-year period (1990–2019), offering insights into the temporal and geographical characteristics of Arctic PMOA.

### 2.1 The aerosol-climate model ECHAM-HAM


The atmospheric simulations for this study are performed with the global state-of-the-art aerosol-climate model system ECHAM-HAM (version ECHAM6.3-HAM2. Tegen et al., 2019). ECHAM simulates atmospheric circulation and dynamics while aerosol microphysics and transport are modelled by the Hamburg Aerosol Module (HAM Stier et al., 2005; Zhang et al., 2012), which is online coupled to ECHAM. HAM is based on the M7 aerosol model (Vignati et al., 2004) that represents aerosols

as soluble or insoluble modes, comprising seven log-normal classes that fall into a size spectrum of four categories depending on the particle radius (r): nucleation (r $\leq$ 0.005 $\mu$m), Aitken (0.005 $\mu$m$<$r$\leq$ 0.05 $\mu$m), accumulation (0.05 $\mu$m$<$r$\leq$ 0.5 $\mu$m) and coarse modes (r$>$ 0.5 $\mu$m). The model includes several aerosol species such as sulphate (SO$_4$), organic carbon (OC), black carbon (BC), mineral dust (DU) and sea salt (SS). Leon-Marcos et al. (2024) implemented PMOA in the model as an additional tracer in the accumulation size mode. PMOA emissions are based on the premise that marine organic matter is co-emitted with

SS as sea spray. Hence, the mass $(M)$ of sea spray can be calculated as $M(seaspray) = M(PMOA) + M(SS)$. Consequently,



the estimated emission mass flux of PMOA groups ($\text{PMOA}_{massflux}$) can be derived from that of sea salt ($\text{SS}_{massflux}$), given the fraction that organics represent of sea spray:

$$\text{PMOA}_{massflux}(i) = \frac{\text{SS}_{massflux} * \text{OMF}_i}{1 - \text{OMF}_i}, \tag{1}$$

where $\text{SS}_{massflux}$ in the model is calculated based on the Long et al. (2011) source function, considering a surface temperature correction in accordance with Sofiev et al. (2011). $\text{OMF}_i$ is the organic mass fraction of each biomolecule group $i$ obtained from the parameterization OCEANFILMS (Organic Compounds from Ecosystems to Aerosols: Natural Films and Interfaces via Langmuir Molecular Surfactants Burrows et al., 2014) that has been recently included as part of the PMOA implementation.

## 2.2 Source representation of primary marine organic aerosol

The OCEANFILMS parameterization represents the transfer of marine organics to the atmosphere (Burrows et al., 2014). It estimates the organic mass fraction in nascent sea spray aerosols of various organic groups. The scheme is based on the Langmuir isotherm model, which represents the differential absorption of organics at the bubble surface. Each group is characterized by distinct physico-chemical properties that will determine their transfer to the aerosol phase. The aerosolization of these marine organics occurs in a chemoselective manner, in which the compounds with higher surface affinity, such as lipids, are preferably transferred. Other molecules that possess a lower surface affinity, such as proteins, polysaccharides, humic and processed compounds, are also considered in OCEANFILMS. However, only three groups are included in this study: lipids, polysaccharides, and protein-like mixtures. Excluding the other groups that originate from the recalcitrant portion of DOC in seawater has a negligible effect on the aerosol organic mass fraction (Burrows et al., 2014). A more extensive explanation of the model characteristics and the methodology employed to compute the biomolecules can be found in Leon-Marcos et al. (2024).

### 2.2.1 Ocean biomolecule concentration

As lower boundary conditions for the OCEANFILMS scheme in ECHAM-HAM, we use simulation results from the Regulated Ecosystem Model (REcoM, version 3) coupled to the general circulation and sea-ice Finite VolumE Sea-ice Ocean Model (FE-SOM, version 2.1) (Gürses et al., 2023). FESOM-REcoM simulates globally the ocean dynamics and marine biogeochemistry, respectively. REcoM includes two types of phytoplankton and zooplankton, as well as nutrients, dissolved and particulate organic matter, and debris (Oziel et al., 2025). Phytoplankton metabolism, such as carbon exudation, is controlled by non-linear limiting functions based on the intracellular nitrogen-to-carbon ratio (Geider et al., 1998; Schourup-Kristensen et al., 2014). The FESOM uses an unstructured grid, which allows for higher resolution in dynamically active regions like the Arctic. For the present investigation, we utilize monthly values of the FESOM-REcoM simulations, which were interpolated to a regular grid



with a horizontal resolution of 30 km. Furthermore, a volume-weighted average over the top 30 meters of the water column is used to represent the marine tracers at the ocean surface.

Based on REcoM model tracers, Leon-Marcos et al. (2024) developed a closure approach to simulate the most abundant biomolecule groups in seawater. The approach considers the main products of dissolved organic carbon exuded by phytoplank-
ton ($DOC_{phy\_ex}$). This fraction of the DOC is apportioned into the contribution of different biomolecule groups in addition to a residual. The main biomolecules in seawater considered are dissolved carboxylic acidic containing polysaccharides ($PCHO_{sw}$), dissolved combined amino acids ($DCAA_{sw}$) and polar lipids ($PL_{sw}$). Any compound that does not belong to the previously mentioned groups is attributed to the residual.

The ocean concentrations of the biomolecular groups are calculated using different methods. PCHO is computed online as
a tracer in the current REcoM model (Leon-Marcos et al., 2024) representing a significant portion of exuded carbon (63 %, (Engel et al., 2004; Schartau et al., 2007)). $PCHO_{sw}$ aggregation product is also computed as sink term and considered an additional model tracer (Transparent Exopolymer Particles (TEP)).

On the other hand, $PL_{sw}$ is calculated offline and accounts for a small fraction of $DOC_{phy\_ex}$ (5 %). The calculation for the $PL_{sw}$ group incorporates the phytoplankton carbon exudation rate over a short timescale of a few days, accounting for its role
as a semi-labile compound. Lastly, $DCAA_{sw}$ is estimated as a fraction of modelled $PCHO_{sw}$. This fraction refers to the ratio derived from analogous compounds of these two groups in seawater samples. As measurements are incapable of distinguishing between biomolecule sources in the ocean, the computed $DCAA_{sw}$ concentration may encompass other sources besides phytoplankton carbon exudation. Hence, as $PCHO_{sw}$ corresponds to the semi-labile group in the ocean, with turnover periods spanning from months to years, the calculated $DCAA_{sw}$ will also be included in this portion. The offline precalculated
ocean concentrations of the three biomolecule groups are finally provided as input files for the marine emission scheme in the ECHAM-HAM model.

### 2.2.2 Experimental model setup

The simulations of PMOA were conducted with ECHAM-HAM for the thirty-year period spanning from 1990 to 2019, for
which also the FESOM-REcoM model output is available. The biomolecule ocean concentration serves as boundary condition for ECHAM-HAM, as explained in the previous section. The model was run at a T63 horizontal resolution, equivalent to approximately, $180 \times 180$ km, with 47 vertical layers. A spin-up time of one year and an output frequency of 12 hours is considered. The simulations are performed in nudged mode with ECMWF ERA-Interim and ERA-5 reanalysis data. The sea ice concentration (SIC) and sea surface temperature (SST) boundary conditions are from the Atmospheric Model Intercomparison
Project (AMIP Taylor et al., 2000).



## 2.3 Methodological challenges analyzing marine biomolecules in the Arctic

Analyzing ocean biomolecules in the Arctic presents specific challenges. Although REcoM simulates marine biogeochemistry beneath sea ice, under-ice production does not contribute to sea spray emissions, since ice cover prevents bubble bursting at the surface. This mismatch complicates linking modelled under-ice biomolecule concentrations to aerosol sources. Therefore, when characterizing ocean biomolecule levels relevant for sea spray production, we exclude grid cells covered by ice—where primary marine organic aerosol (PMOA) emissions are unlikely. Hence, a sea ice mask was applied before calculating the biomolecule ocean concentration over the Arctic. For simplicity, we only consider open ocean conditions (SIC<10 %, (Arrigo et al., 2008)). Nevertheless, sea spray emissions via bubble bursting arise not only over ice-free ocean waters but also within the marginal ice zone and inside the Arctic sea ice pack from open leads and melt ponds (as special features of sea ice) (Leck and Bigg, 2005b; Willmes and Heinemann, 2015; Zhang et al., 2018; Rolph et al., 2020). Note that this mask is only used to average the above parameters over the Arctic and does not apply to the use of the biomolecule ocean concentrations as bottom boundary condition within the ECHAM-HAM simulations. Additionally, for a more profound understanding of the particularities within the Arctic Ocean, we conducted a detailed, separate analysis of the main Arctic seas, as illustrated in Fig. 1.

In this study, Arctic trends were assessed using the non-parametric Mann–Kendall test and the Theil–Sen slope estimator. For marine variables, we must also consider that the production under ice is present. However, when computing the trends of ocean biomolecule concentration, we did not apply the ice mask described above. Excluding under-ice production led to inconsistent and unrealistic trend patterns because interannual and seasonal variability of sea ice, especially near the ice edge, strongly influences marine production. This likely reflects differing bloom dynamics in the marginal ice zone versus fully open-ocean areas. Hence, we estimated the changes in the marine biomolecules by computing maximum trends of likely ice-free regions within the Arctic. To achieve this, we excluded areas overlapping the seasonal minimum sea ice concentration. This ensures that potentially open-water regions, where marine organic emissions could occur over the 30-year period, are taken into account. Finally, trends of emission mass fluxes and aerosol concentration of sea salt aerosol and PMOA modelled by ECHAM-HAM are also analysed in Section 4.





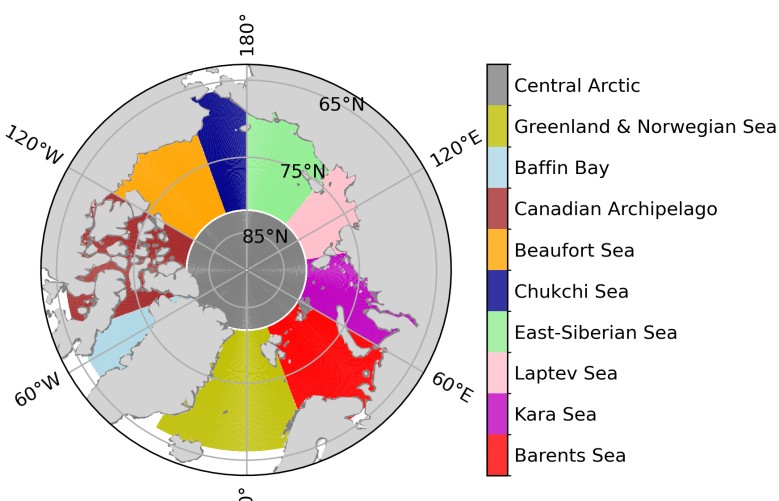

**Figure 1.** Map of Arctic Ocean subregions considered in the present study. Lateral boundaries were defined following the oceanic region definitions by Nöthig et al. (2020) and Randelhoff et al. (2020), whereas latitudinal limits were modified and extended to uniformly cover 66 °N–82 °N for all regions except the Central Arctic (82 °N–90 °N).

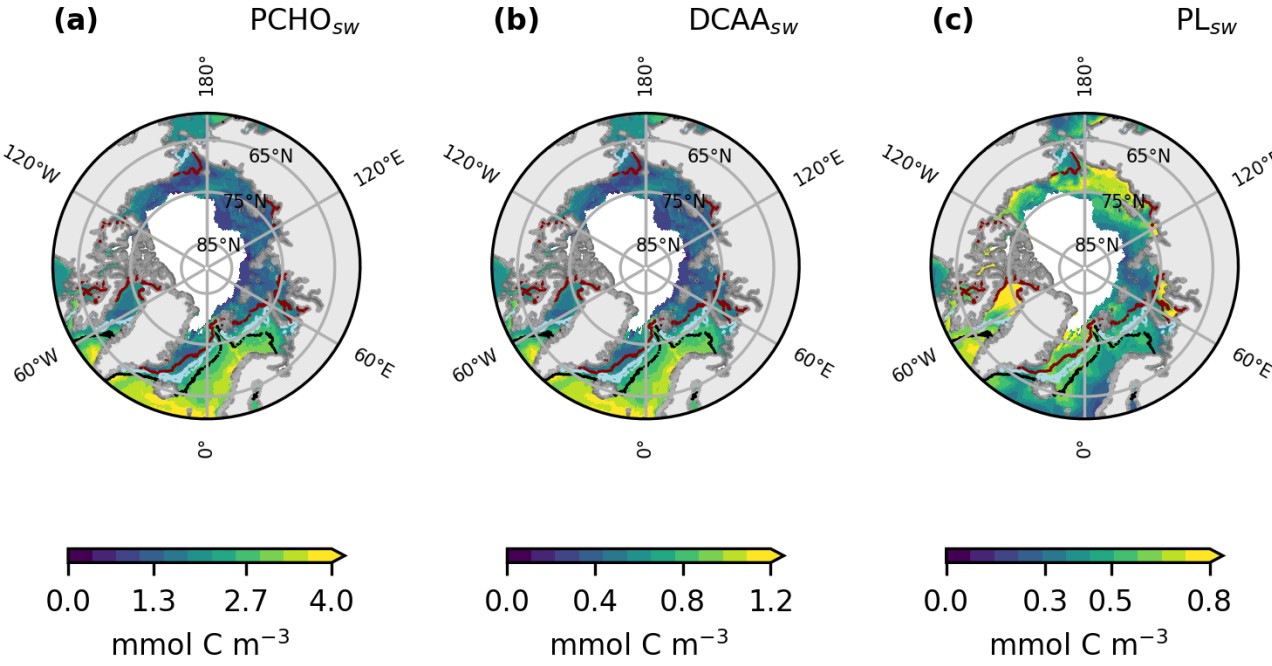

**Figure 2.** Maps of averaged ocean carbon concentration of (a) $PCHO_{sw}$, (b) $DCAA_{sw}$, and (c) $PL_{sw}$ as a multiannual mean spanning May–September for the period 1990-2019. The black, blue and red lines depict the ice edge, defined as the contour of 10 % sea ice concentration for May, July and September, respectively.



## 3   Results and Discussion

### 3.1   Geographical distribution of marine biomolecule groups

Biomolecule ocean concentration is shown in Fig. 2 for the compounds simulated in the present study as multiannual average over the period 1990–2019. In terms of carbon contribution, $PCHO_{sw}$ dominates in seawater, with a mean concentration over the Arctic circle of 1.4 mmol C m$^{-3}$, followed by $DCAA_{sw}$ (0.4 mmol C m$^{-3}$) and $PL_{sw}$ (0.3 mmol C m$^{-3}$). The distribution of $PCHO_{sw}$ and $DCAA_{sw}$ (Fig. 2a, b) present a nearly identical geographical distribution, since the latter was computed as a fraction of simulated $PCHO_{sw}$. In contrast to $PL_{sw}$, spatial patterns are rather distinct (Fig. 2c). For instance, notable greater concentrations are seen in the Norwegian Sea and North Atlantic compared to the central Arctic and vice versa for the semi-labile and lipid group, respectively. These differences also vary along the year. Hence, a description of the seasonal particularities of regions within the Arctic Ocean that determine the distribution of the biomolecules is provided further below. The differing geographical distribution of the groups is determined by the production or loss mechanisms considered in the biomolecules' computation. $PCHO_{sw}$ represents the largest fraction of phytoplankton exuded DOC. It quickly aggregates to form TEP, which is considered a loss term in the online simulation of $PCHO_{sw}$ by REcoM. This is the reason for the prominent differences in the Arctic Ocean biomolecule concentration compared to $PL_{sw}$ group (see Fig. 2a).

### 3.2   Seasonality of marine biomolecule groups

The biomolecule quantities have a pronounced seasonality in the polar regions (Fig. 3). When light limitation decreases at the end of the polar night, phytoplankton begins to grow. Figure 3a illustrates the seasonal cycle of the ocean carbon concentration of the biomolecules averaged over the Arctic Ocean from 1990 to 2019 considering solely sea ice free ocean conditions (SIC<10 %, Arrigo et al. (2008)). The seasonal patterns vary among the organic groups. $PCHO_{sw}$ and $DCAA_{sw}$ ocean concentration rise sharply until May, whereas $PL_{sw}$ peaks a month later. The presence of all biomolecules is high from April to October, with a gradual decrease after their peak in early summer. $PCHO_{sw}$, as the major extracellular product of phytoplankton in seawater, its concentration remains higher than the $DCAA_{sw}$ and $PL_{sw}$ groups across the months. Maximum concentration of $PCHO_{sw}$, $DCAA_{sw}$ and $PL_{sw}$ are 5.4 ± 1.5, 1.6 ± 0.5 and 0.9 ± 0.3 mmol C m$^{-3}$, respectively.

The dominance of the biomolecules in the ocean during spring and summer occurs in response to the higher phytoplankton carbon concentration in the water for this period. After the rapid consumption of available nutrients during the phytoplankton growth, the bloom decays mostly due to nutrient depleted conditions. Among the modelled phytoplankton groups, diatoms contribute to the majority of the exuded DOC in the Arctic, especially during the early stage of the bloom (Fig. A1).

The OMF in nascent aerosol shows a similar seasonal pattern, with the highest contributions in spring and summer (Fig. 3b). However, the OMF of the aerosol species ($PCHO_{aer}$, $DCAA_{aer}$ and $PL_{aer}$) do not behave as their precursors in the ocean. $PCHO_{aer}$ has the lowest OMF, followed by $DCAA_{aer}$ and $PL_{aer}$. As previously explained, the high surface affinity of lipids, positions $PL_{aer}$ as the major contributor to marine organic aerosol during months with high biological productivity. Values are as high as 0.4 ± 0.05. OMF for $PL_{aer}$ are at least one to two orders of magnitude higher than for $PCHO_{aer}$ and $DCAA_{aer}$,



respectively. Whereas $PCHO_{aer}$ and $DCAA_{aer}$ remain within $10^{-3}$ and $10^{-2}$ throughout the year (note that $PCHO_{sw}$ has the
lowest surface affinity), $PL_{aer}$ decreases to negligible values as the $PL_{sw}$ concentration in the ocean approaches almost zero in
winter months (Fig. A1).

Note that we averaged the ocean concentrations and OMF over the whole Arctic region, which does not represent the spatial
particularities and seasonality of all subregions within the Arctic circle (Fig. 2). Ocean marine productivity in REcoM is lim-
ited by either light or nutrient availability, which is influenced by physical features like advection, mixing, stratification, sea
ice and ocean temperature (Schourup-Kristensen et al., 2018). Hence, $PL_{sw}$ concentration in Fig. 2c shows different patterns
for various sites in the Arctic (see Fig. 3c and with a large variation among regions between May and August). In the present
study, we provide an overview of the seasonal climatology of $PL_{sw}$ as the most relevant biomolecule for the aerosol OMF.

The seasonality of $PL_{sw}$ has a close similarity to that of the phytoplankton carbon concentration (Fig. A2a). The phytoplankton
bloom starts when light limitation is alleviated under nutrient enriched conditions. Consequently, $PL_{sw}$ ocean concentration
growth for the Arctic seas starts between March and May, reaching the maxima in May, June, or July. The patterns in the
seasonal climatology are determined by the particularities of each subregion. Conversely, total OMF exhibits lower variability
than $PL_{sw}$ concentration in seawater, yet their seasonality aligns closely with OMF reaching maximum values between 0.37
and 0.45.(Fig. 3d)

Sea ice is a controlling factor in the initiation of the bloom (Ardyna and Arrigo, 2020), as well as the magnitude of the
biomolecule production. For instance, in the Central Arctic, a less prominent late bloom shifts the initiation of phytoplankton
carbon release to May (see Fig. 3c). The maximum values are seen in August, with $PL_{sw}$ concentration and Total OMF values
over $0.8\,\mathrm{mmol\,C\,m^{-3}}$ and 0.4, respectively. This region is characterized by the highest SIC. Therefore, light is the most lim-
iting factor here (Schourup-Kristensen et al., 2018) as sea ice persists and only partially retreats by mid-summer (Fig. A2b).
Furthermore, low nutrient availability is also typical of the central Arctic, which keeps the net primary production (NPP) low.
For the Greenland, Norwegian, and Barents Seas, the $PL_{sw}$ ocean concentration and OMF are restricted to smaller values
compared to the Arctic mean. Quantities are less relevant for the Barents Sea. These regions are strongly influenced by the
lateral transport of nutrients from the North Atlantic Ocean (Harrison et al., 2013). Furthermore, they have typically lower sea
ice coverage compared to all Arctic subregions and the ice tends to be thinner and younger. Hence, light is not a strong limiting
factor except for the Western Greenland Sea and northern Barents Sea areas, in which the biological activity is enhanced when
sea ice melts (Fig. 2).

Interestingly, the Chukchi Sea, which is also largely influenced by the lateral nutrient supply from the Pacific Ocean through
the Bering Strait (Walsh et al., 1989), present about 1.5 greater values than the Atlantic Ocean neighbouring waters. Unlike the
Nordic seas, which have nearly ice free conditions throughout the year, the Chukchi is fully ice covered during winter, being a
remarkable difference between these regions.
In addition to the Chukchi Sea, the Russian shelf, Beaufort Sea and Canadian Archipelago are also seasonally sea ice covered
(Fig. A2b). $PL_{sw}$ concentration and OMF lay above the Arctic average for these areas. Values extend up to $1.2\,\mathrm{mmol\,C\,m^{-3}}$ and
0.4 for the ocean and aerosol variables, respectively. In the coastal zones of these regions, the sea ice cracks and melts, which, in
combination with local factors, rapidly triggers the ocean marine primary production. In addition, the Eastern Siberian, South-



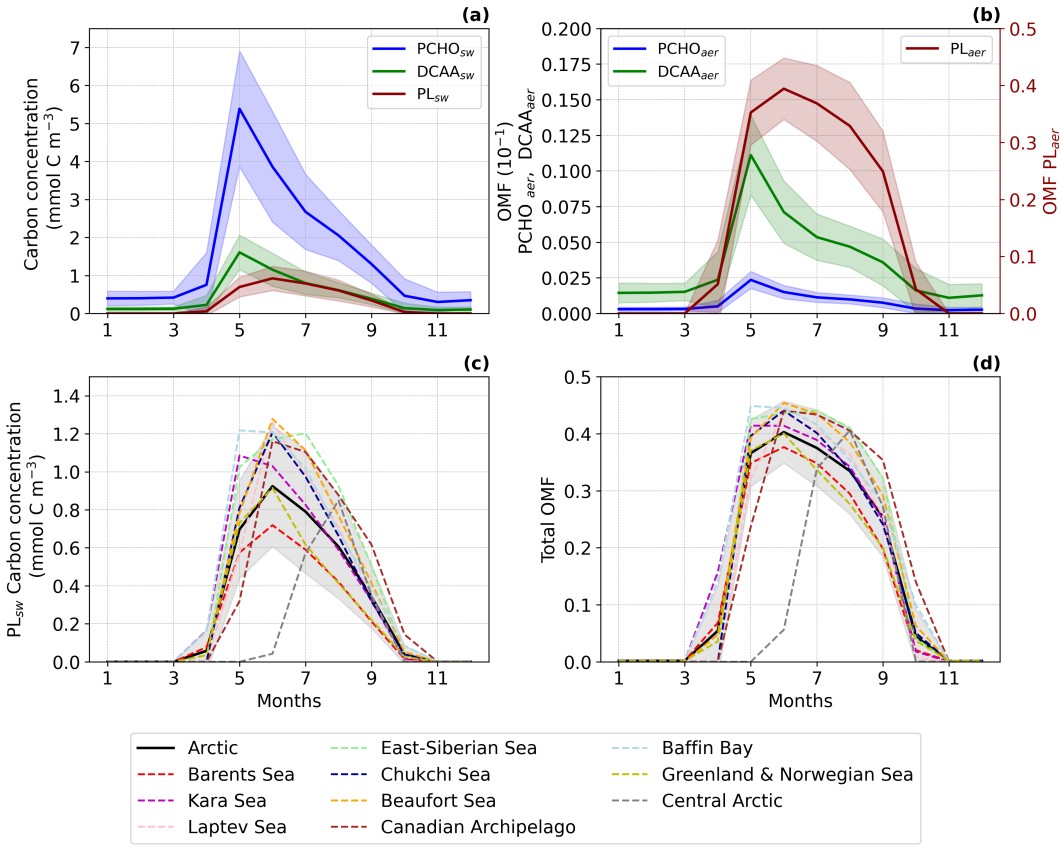

**Figure 3.** Seasonal climatology of (a, c) the ocean carbon concentration for $PCHO_{sw}$, $DCAA_{sw}$ and $PL_{sw}$ and, (b, d) offline simulation of organic mass fraction (OMF) in nascent aerosol from OCEANFILMS for $PCHO_{aer}$, $DCAA_{aer}$ and $PL_{aer}$ for the period 1990–2019 and sea ice free ocean conditions (SIC<10 %, (Arrigo et al., 2008)) averaged over the Arctic (a, b). Total OMF refers to the aggregated organic mass fraction of all biomolecules. Thick colour lines show the average over the Arctic circle (66 °N–90 °N) and, dashed lines (c, d) illustrate the seasonality of all seas within the Arctic Ocean (Fig. 1). The shaded area represents the spatial standard deviation of the long-term monthly mean.

ern Beaufort, Laptev and Kara Seas are characterized by the large influence of land and higher concentrations of biomolecules

are attributed to the riverine supplies of nutrients (Miquel, 2001; Wang et al., 2005; Karlsson et al., 2011; Oziel et al., 2025). Note that ice-edge blooms and high nutrients near shore in ice-free conditions are the sites with the highest $PL_{sw}$ production (Fig. 2c), suggesting that its spatial distribution is highly sensitive to sea ice dynamics.

## 3.3 Patterns of PMOA emissions

Like the biomolecule concentration in the ocean, PMOA emission mass flux also follows a specific seasonality in the Arctic (Fig. 4). Sea ice strongly influences marine aerosols by influencing ocean bioactivity and limiting sea spray emissions via





bubble bursting. As a result, marine aerosol emission mass fluxes are expected to increase as sea ice melts. In the next sections we present the geographical distribution of the emissions as well as their seasonality in contrast to the main emission drivers.

### 3.3.1 Geographic distribution

Figure 4 shows the geographical distribution of mean emission flux for each group for the winter months January-February-March and summer July-August-September. During the polar night, biomolecules in the Arctic Ocean remain very low (Fig. 3a–c). Hence, weak emission fluxes are reported in winter with a total PMOA flux of $1.4 \times 10^{-3}$ ng m$^{-2}$ s$^{-1}$. The minimum in marine emissions in winter is accompanied by the maximum sea ice concentration for the season. Marine aerosols are confined to the North Atlantic and Pacific oceans, where high winds promote elevated sea spray emissions. Nonethe-

less, PCHO$_{aer}$ and DCAA$_{aer}$ (Fig. 4a, b) still contribute over the southern Arctic waters (Greenland and Norwegian Seas), with emissions as high as 0.04 ng m$^{-2}$s$^{-1}$. On the other hand, PL$_{aer}$ average flux (Fig. 4c) is negligible for this period ($2.2 \times 10^{-6}$ ng m$^{-2}$s$^{-1}$) whereas the other two groups dominate. The mean values for PCHO$_{aer}$ and DCAA$_{aer}$ are $2.5 \times 10^{-4}$ and $1.2 \times 10^{-3}$ ng m$^{-2}$s$^{-1}$, respectively.

In contrast to winter, summer fluxes are moderate for the North Atlantic and Pacific Oceans (Fig. 4d–f). Nevertheless, mean

quantities are greater over the Arctic compared to winter months with values of $7.1 \times 10^{-4}, 3.4 \times 10^{-3}$ and $1.8 \times 10^{-1}$ ng m$^{-2}$s$^{-1}$ for PCHO$_{aer}$, DCAA$_{aer}$ and PL$_{aer}$ respectively. As the phytoplankton bloom sets in during the melting season, marine organic aerosols become relevant and expand northward over the Norwegian, Greenland, Baltic, and Chukchi Seas. Unlike winter, the minimum in sea ice for the period leads to a maximum in organic emissions (0.18 ng m$^{-2}$s$^{-1}$). Among the aerosol groups, PL$_{aer}$ contributes to most of the organic mass fraction in aerosols. Compared to the other groups, the contribution of PL$_{aer}$ is

widely spread across the Arctic seas, being the specie with the strongest increase from winter to summer. Note that the marine aerosol contribution varies per species and regions within the Arctic circle (Fig. 4). A comprehensive analysis of the seasonal characteristics of marine emissions is presented further below.

To study how total marine emissions in the Arctic have changed over time, we calculated the average of the accumulated fluxes and burden of marine aerosols for the first and second half of the simulated period (Table 1). As expected, PL$_{aer}$ accounts for

the majority of PMOA and represents 2.4 % of total emitted SS for the 30-year period. Conversely, PCHO$_{aer}$ and DCAA$_{aer}$ make up to 0.07 % and 0.02 %, respectively. Note that SS emissions includes the accumulation and coarse modes as a model output variable, while PMOA is emitted in the accumulation mode only. Hence, the actual PMOA/SS fraction may be higher if we considered the accumulation mode only.

For the 15-year periods, a noticeable increment in the emissions is seen for all species (Table 1). PCHO$_{aer}$ presents the largest

augment, with an 19.3 % increase from 1990–2004 to 2005–2019. Conversely, DCAA$_{aer}$, PL$_{aer}$ and SS growth is less strong, with values of 12, 13.9 and 10.6 %, respectively. In our model, burden values also rise, although not as significant as the changes in emissions. For PCHO$_{aer}$, the positive variations in the burden is also high (6.8 %) in contrast to a lower increase in DCAA$_{aer}$ and PL$_{aer}$ (4.5 and 4.2 %). This indicates that an increment in the aerosol sources will have a positive effect on the column burden. Similarly, the aerosol removal increases accordingly (Table 1). Wet deposition in stratiform clouds and

in-cloud processes are the main processes that govern the loss of marine organics. For PCHO$_{aer}$, DCAA$_{aer}$ and PL$_{aer}$ the





percent of increase is about twice larger than for the burden (13.9, 8.9 and 9.7 %, respectively). In contrast, SS loss changed from the first half to the second half of the period is only slightly larger than the burden increase (8.8 %). Hence, estimated PMOA residence time in the atmosphere shortened for all species from 4 to 6 %. The noticeable differences between $PL_{aer}$ and $DCAA_{aer}$ and $PCHO_{aer}$ are primarily attributed to the variations in the geographical distribution (Fig. 4) and seasonality

of aerosol fluxes (see next section) in the Arctic.

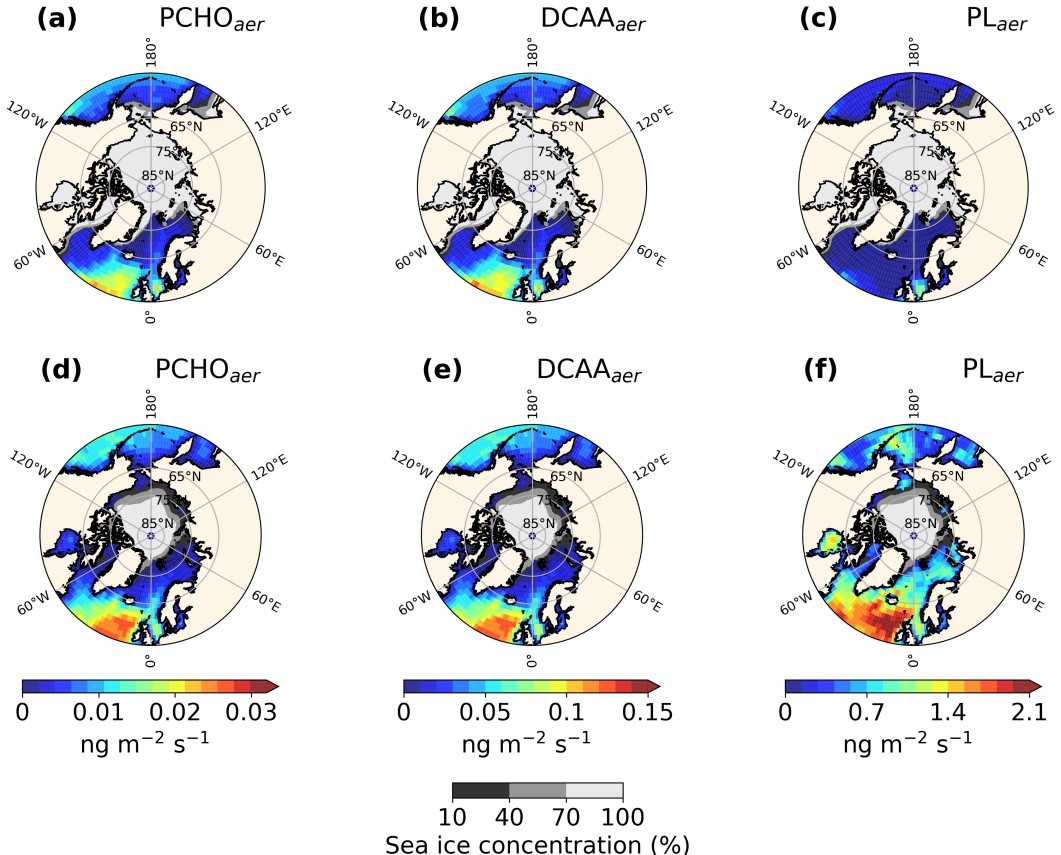

**Figure 4.** Maps of Surface emission mass flux of (a, d) $PCHO_{aer}$, (b, e) $DCAA_{aer}$ and (c, f) $PL_{aer}$ for the Arctic averaged over (a–c) January-February-March and (d–f) July-August-September for the simulated period 1990–2019.

### 3.3.2 Seasonality of sea spray aerosol and emission drivers

Wind is the main driver for SS emission flux. This is followed by the linear relationship with open ocean fraction (1-SIC) and a correction factor based on SST. Nonetheless, the relevance of these drivers could vary for different Arctic subregions. To disentangle the relative influence of sea spray emission drivers in ECHAM-HAM model, in this section, we discuss the

seasonality of SIC, SST and 10-m wind speed in relation to sea salt fluxes and their impact on the PMOA emissions in the Arctic (Fig. 5).



**Table 1.** Average of accumulated emission flux, burden, and deposition of marine aerosol particles over the Arctic for 15-year periods.

| Emission mass (Tg yr$^{-1}$) | | | |
|---|---|---|---|
| | 1990–2004 | 2005–2019 | **1990–2019** |
| PCHO$_{aer}$ | $3.2 \times 10^{-4}$ | $3.8 \times 10^{-4}$ | **$3.5 \times 10^{-4}$** |
| DCAA$_{aer}$ | $1.7 \times 10^{-3}$ | $1.8 \times 10^{-3}$ | **$1.7 \times 10^{-3}$** |
| PL$_{aer}$ | $5.1 \times 10^{-2}$ | $5.8 \times 10^{-2}$ | **$5.5 \times 10^{-2}$** |
| SS | $2.2 \times 10^{0}$ | $2.4 \times 10^{0}$ | **$2.3 \times 10^{0}$** |
| Burden (Tg) | | | |
| PCHO$_{aer}$ | $1.7 \times 10^{-6}$ | $1.8 \times 10^{-6}$ | **$1.8 \times 10^{-6}$** |
| DCAA$_{aer}$ | $8.1 \times 10^{-6}$ | $8.5 \times 10^{-6}$ | **$8.3 \times 10^{-6}$** |
| PL$_{aer}$ | $1.5 \times 10^{-4}$ | $1.5 \times 10^{-4}$ | **$1.5 \times 10^{-4}$** |
| SS | $2.6 \times 10^{-3}$ | $2.8 \times 10^{-3}$ | **$2.7 \times 10^{-3}$** |
| Aerosol deposition ( Tg yr$^{-1}$) | | | |
| PCHO$_{aer}$ | $4.7 \times 10^{-4}$ | $5.3 \times 10^{-4}$ | **$5 \times 10^{-4}$** |
| DCAA$_{aer}$ | $2.3 \times 10^{-3}$ | $2.5 \times 10^{-3}$ | **$2.4 \times 10^{-3}$** |
| PL$_{aer}$ | $5.8 \times 10^{-2}$ | $6.4 \times 10^{-2}$ | **$6.1 \times 10^{-2}$** |
| SS | $2.0 \times 10^{0}$ | $2.2 \times 10^{0}$ | **$2.1 \times 10^{0}$** |

Figure 5a. shows the average 10-m winds for the Arctic subregions. In the neighbouring North Atlantic waters, Baffin Bay and Barents and Chukchi seas, winds follow the seasonal meteorological conditions, with intensified velocities in the winter months. For the inner Arctic seas, patterns are more heterogeneous. The Central Arctic, Kara Sea, Beaufort Sea and Canadian
Archipelago do not present a pronounced seasonality, whereas, the Laptev and East-Siberian winds tend to be higher in summer.

Open ocean fraction follows a similar seasonality for all Arctic subregions, as sea ice shrinks through the summer and refreezes during winter (Fig. 5b). Before the onset of the melting season, the Greenland and Norwegian Seas present the highest open water fractions, nearing 80 %. The Barents Sea ranks next, with values between 60 and 70 %. In contrast, the Central Arctic
experiences only a modest summer SIC reduction, maintaining an open water fraction generally below 10 % throughout the year. Other subregions display values between 10 % and 60 % during the year's first five months, followed by a summer increase. Most of these regions experience approximately 40 % sea ice loss, with the most pronounced reductions occurring in September. In summer, the Beaufort Sea surpasses 65 % open water, whereas the Canadian Archipelago, East Siberian, and Chukchi Seas approach 80 %. The Chukchi Sea exhibited the most pronounced transformation, with nearly a 70 % increase
compared to winter. Lastly, the Arctic's rising SST (Fig. 5c) corresponds to the increase in the fraction of open ocean. In this case, the warmest temperatures occur in parallel to the lowest sea ice cover. The amplitude of SST for each region varies between one and two degrees Celsius and is similar to that seen in Fig. 5b. Nevertheless, for the Chukchi Sea, seasonality is more prominent given the strong changes in SIC. Similarly, the Greenland, Norwegian, and Barents seas show strong seasonal patters; however, temperatures are warmer and remain positive throughout the year. Overall, SST ranges between -2 to 6 °C.
Within this temperature range, the Sofiev et al. (2011) SST correction factor used in SS model representation remains relatively similar for the accumulation mode particles. Therefore, in this case, SST has a smaller effect on marine emissions.

Sea salt aerosol seasonality shows very similar patterns to the 10-m wind speed for the Barents and Greenland and Norwegian





Seas, in which the emissions are the largest in the Arctic Ocean (Fig. 5d). Values steadily decrease from January to June, with a smooth increase until October. Note that the average open water fraction remains larger than 80 %. However, the variations

in the Barents Sea are less pronounced as it partially freezes and hinders the emissions in winter. This is well illustrated when comparing the emission drivers in the Nordic Seas and Chukchi Sea. For the latter, wind strength lays close to that in the Barents Sea; however, the open ocean is twice smaller than in the Barents region. Similarly, for the remaining subregions, the fraction of open water remains lower compared to the Nordic seas throughout the year. Sea spray production commences between May and June, months in which sea ice starts melting. Among the inner Arctic seas, Chukchi Sea has the stronger contributions

in October with high surface winds occurrence ($6.8 \, \mathrm{m \, s^{-1}}$). Following this region, Kara Sea and Baffin Bay have the greatest contributions in September (with the maximum in open ocean fraction) and October (conditioned by a peak in surface winds). Laptev and East-Siberian Sea emissions are close together as a result of contrasting wind and SIC patterns. Weaker emissions are also found in the Beaufort Sea and Canadian Archipelago, strongly controlled by the sea ice cover. Lastly, in the central Arctic, the fluxes are as extremely low despite the presence of stronger than Arctic average winds, although with the smallest

open ocean areas for sea spray occurrence.

Given the cyclic life of phytoplankton blooms, ocean biomolecules and OMF increase during the polar day and sharply decay at the end of the Arctic summer. Consequently, organic aerosol emission fluxes present distinct characteristics compared to SS and among Arctic subregions (Fig. 5e, f). Furthermore, the interannual variability of PMOA groups is stronger during the high productivity season, while SS deviations are larger during winter. The most relevant discrepancies with SS seasonal patterns

are seen in the Barents, Greenland and Norwegian Seas, in which the curve slightly resembles the biomolecule OMF instead (see also Fig. 3d, e). Nonetheless, as a result of stronger SS fluxes, the magnitude of the organic aerosol emissions remain larger in the Nordic Seas compared to other Arctic subregions.

As previously discussed, $\mathrm{PL}_{aer}$ and $\mathrm{PCHO}_{aer}$+$\mathrm{DCAA}_{aer}$ present different seasonality and abundance in the ocean and atmosphere (see also Fig. 3b). For instance, $\mathrm{PL}_{aer}$ has notable contributions during the Arctic summer, whereas, the semi-labile

compounds also contribute outside the bloom period (see also Fig. 4a, b). Note that $\mathrm{PCHO}_{aer}$+$\mathrm{DCAA}_{aer}$ emissions have a bimodal distribution for the Nordic seas, with a global maximum in May. For these areas, the contributions drop in July to its minimum, associated with the lowest wind velocities. Emissions rise to a second maximum in September, triggered by the SIC decline in summer. This peak later in summer is less prominent in the Barents Sea compared to the Norwegian and Greenland Seas. Values continue to decay until November, with a moderate increase during the polar night, a period in which $\mathrm{PL}_{aer}$

production is absent. Fig. 5f shows that the $\mathrm{PL}_{aer}$ emission fluxes have a similar pattern to that of $\mathrm{PCHO}_{aer}$+$\mathrm{DCAA}_{aer}$ for the Greenland and Norwegian Seas. Interestingly, in the Barents Sea, $\mathrm{PL}_{aer}$ does not have a bimodal pattern and persists high from May to June, corresponding to the $\mathrm{PL}_{aer}$ OMF.

Notably, weaker emissions of marine biomolecules occur in the other Arctic subregions due to weaker sea salt fluxes. As sea spray occurrence is strongly affected by sea ice cover, organic aerosols become more relevant towards the end of the melting

season. Hence, organic emissions peak through July to September to decline to values near to zero throughout the winter. For these regions, PL emission seasonality has more similarities to that of $\mathrm{PCHO}_{aer}$+$\mathrm{DCAA}_{aer}$. Nevertheless, the latter often reach their seasonal peak ahead of $\mathrm{PL}_{aer}$. The Chukchi Sea has the highest emissions, followed by Kara Sea, Baffin Bay, East-



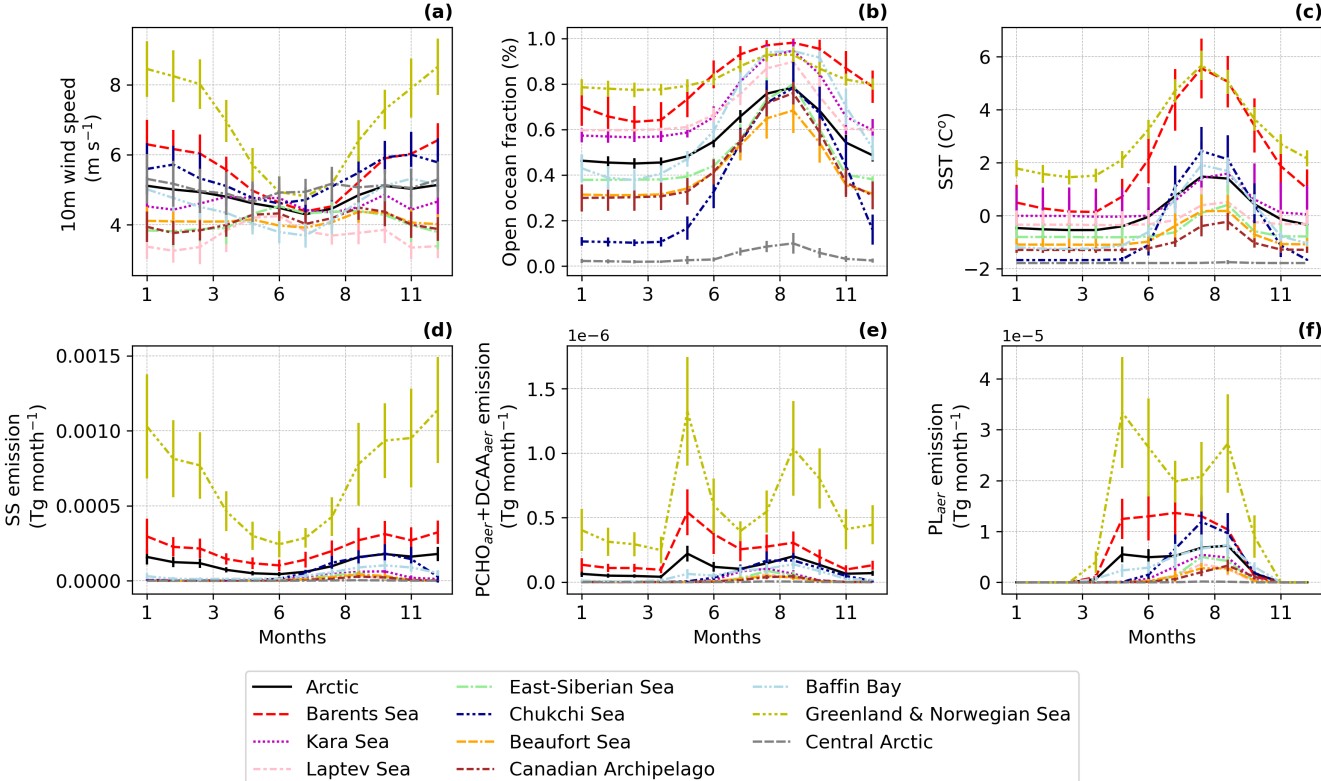

**Figure 5.** Seasonal climatology of (a) 10m Wind speed, (b) open ocean fraction, (c) SST and emission fluxes of (d) SS, (e) $PL_{aer}$ and (f) $PCHO_{aer} + DCAA_{aer}$ for the period 1990–2019 simulated by ECHAM-HAM model averaged over the Arctic and all seas within the Arctic Ocean (Fig. 1). The error bars indicate the multiannual standard deviation.

Siberian and Laptev seas. The distinction in magnitude of emissions among these seas is primarily triggered by the summer sea ice melt. Note that, compared to sea salt, the slopes, and the smoothness of the curves vary for all marine species. These

distinguishable characteristics evidence the effect of marine biological activity in determining the patterns in the emission seasonality.

Modelled marine emission patterns in the Arctic are the result of a combination of four main controlling factors, surface winds, open-ocean grid cell fraction, SST and marine productivity. The strong power law dependency of SS on wind speed (Long et al., 2011) produces significantly higher values for slightly stronger winds (e.g., North Atlantic Ocean in contrast to the

Baltic Sea). Nevertheless, the high bioactivity during summer compensates the lower wind-driven sea salt emissions, reaching magnitudes comparable to higher wind speed zones. Therefore, the representation of marine aerosol precursors is essential in polar regions, where seasonality in ocean biological activity and sea ice retreat regulates the organic aerosol emissions during summer.





## 4 Arctic trends

### 4.1 Impact of sea ice retreat on PMOA precursors

To gain deeper insights into how marine biomolecules and their organic contributions to aerosols have evolved under the current Arctic warming, this section examines and discusses observed trends in the Arctic region. Figure 6 shows the trends of the average ocean concentration of $PCHO_{sw}$ and $PL_{sw}$ over July-August-September (summer) in the Arctic region. $DCAA_{sw}$ was not included here as it presents nearly identical characteristics to $PCHO_{sw}$, but lower in magnitude. The minimum SIC for the season overlaps the trends, and it is considered to exclude areas potentially permanently covered by ice. The trends of SIC and net primary production modelled in FESOM-REcoM are also included. In addition, the maximal absolute changes per region for all biomolecules are shown in Fig. 7a. They represent the maximum or minimum values corresponding to the largest fraction of the grid with an increasing or decreasing trend, respectively (Fig. 7b). By using this approach, we ensure that the quantities in Fig. 7a constitute the dominant trend of the region. Note that this analysis was performed considering solely grid cell points where the trends are significant (Mann-Kendall, p-value<0.05; see hatched areas in Fig. 6a, b and Fig. 7c). Figures with the trends for the months April-May-June (spring) are included in the supplement in Fig. A3 and Fig. A4.

$PCHO_{sw}$ concentration (Fig. 6a) increases for most Arctic subregions. The maximum absolute trends remain positive across all subregions for $PCHO_{sw}$ and $DCAA_{sw}$ (Fig. 7a). Most quantities in Fig. 7b, c appear relatively similar for both semi-labile groups. Values in the Canadian Archipelago, East-Siberian and Laptev seas exceed 0.04 and 0.012 $mmol\,C\,m^{-3}\,yr^{-1}$ for each group, respectively. In contrast, the weakest changes are seen in the Baffin Bay. The East-Siberian is the only region with the most consistently increasing trend for all ocean biomolecule concentration (nearly 100 % grid fraction in Fig. 7b). This region is followed by the Barents Sea, both presenting the highest grid fraction with a significant trend (over 19 % in Fig. 7c). Similarly, for the Kara, Laptev and Greenland and Norwegian seas, the trend is significant in an area that represents the 15–17 % of the subregion. However, for these cases, the grid cells with increasing trend range between 80–95 %. Conversely, with roughly 3 % and 6 %, the Beaufort and Chukchi Seas and Canadian Archipelago account for the lowest grid fraction values among all Arctic subregions, respectively (Fig. 7c).

$PL_{sw}$ concentration, on the other hand, increases on the Russian shelf and Beaufort Sea (Fig. 6c). The maximum changes occurred in the Laptev, East-Siberian and Chukchi seas (Fig. 7d). Nevertheless, the density of grid cells with statistically significant trend is small for these regions (under 9 %, Fig. 7c) in contrast to Baffin Bay and Nordic Seas (14–29 %). For the last two cases, $PL_{sw}$ tends to decay, with values ranging from -0.008 to -0.009 $mmol\,C\,m^{-3}\,yr^{-1}$, respectively (Fig. 7d). The strongest decrease is found in the Canadian Archipelago. However, for the Canadian region as well as for Chukchi Sea, the grid fraction with significant trend are as low as 5.8 % and 2.7 %, respectively (Fig. 7c).

In summary, the trends show differing regional characteristics depending on the biomolecule group. For instance, the largest density of model grid points with significant trend for $PL_{sw}$ are found in regions with minor sea ice changes (Baffin Bay, Barents, and Greenland and Norwegian Seas in Fig. 6a, b). On the other hand, for $PCHO_{sw}$ and $DCAA_{sw}$, the inner Arctic seas shares a large grid fraction with a significant trend. Hence, semi-labile biomolecules have predominantly increased in the Arctic, with the most relevant changes found in the Russian shelf. In contrast, $PL_{sw}$ has primarily decreased in some areas,

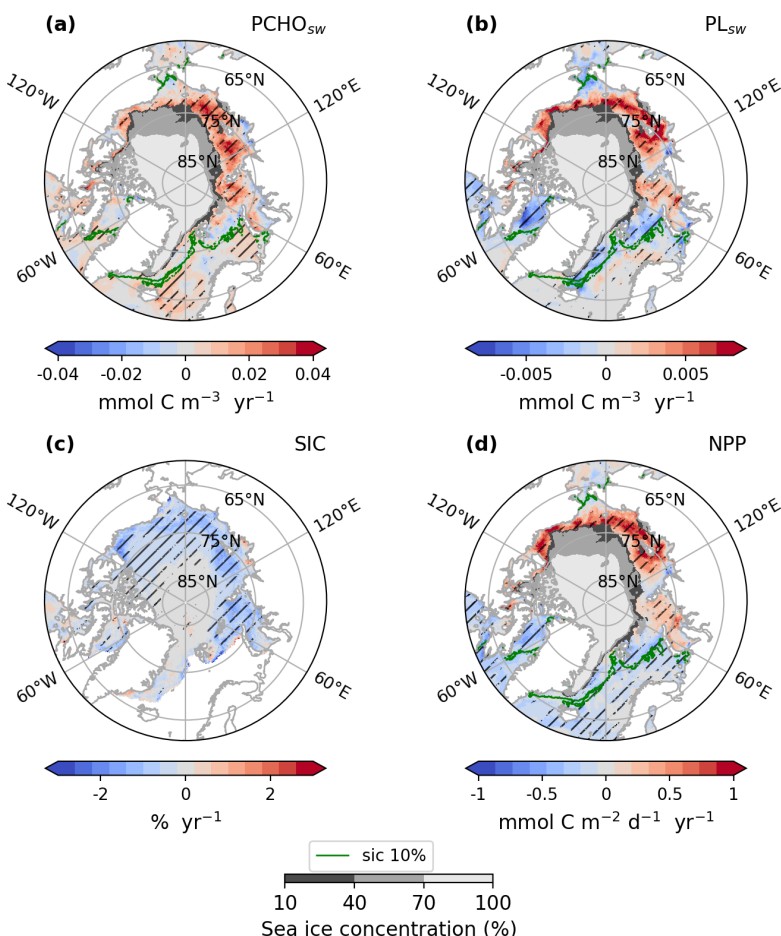

**Figure 6.** Arctic trends of (a) PCHO$_{sw}$ and (b) PL$_{sw}$ ocean concentration, (c) sea ice concentration and (d) net primary production from FESOM-REcoM model for July-August-September of the simulated period 1990–2019. The hatching indicates the areas over which trends are significant (Mann-Kendall test, p-value<0.05). The green contour line depicts the average season 10 % sea ice concentration. The minimum seasonal SIC for the period occurred in September 2012, and it is also represented in shaded gray.



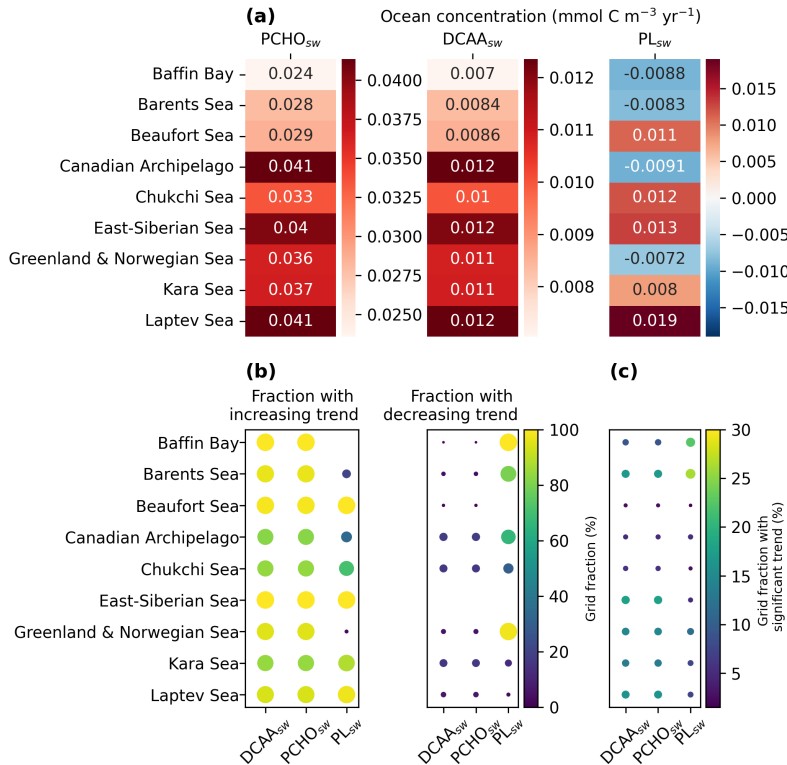

**Figure 7.** Maximum trends for the Arctic subregions in Fig 1 of the (a) biomolecule ocean concentration trend with (b) the highest grid fraction of increasing or decreasing trends for July-August-September of the period 1990–2019. Only cases where the trends are significant (Mann-Kendall test, p-value<0.05) are considered, and (c) illustrate the fraction they represent of each region in terms of percent of grid cell. Values were obtained after applying a mask with the minimum sea ice concentration shown in Fig. 6.

425 with pronounced variations in the Canadian Archipelago and Baffin Bay. Although, the increasing trends, when present, are generally stronger than the negative changes. Note that regions with a strong decline in sea ice generally have a noticeable and statistically significant increase in marine primary production (see Fig. 6c, d). As a result, biomolecules quantities consistently experience an increased in the eastern Arctic subregions during summer.

On the other hand, the wide extension of sea ice cover masks the marine biomolecules that potentially contribute to aerosols during spring (Fig. A3). Hence, in the Russian shelf, the trend is absent for all marine organic groups (see Fig. A4). Nonethe-

430 less, a strong increase in the ocean carbon concentration occurs in the Baffin Bay, Canadian Archipelago and Nordic seas for $PCHO_{sw}$ and $DCAA_{sw}$. The Baffin Bay and Barents Sea absolute maximum are significantly larger than the values later in summer, by about 65 and 48 %, respectively. Similarly, values for the Greenland and Norwegian Seas only slightly decreased in the warmer season. In contrast, for the Canadian area, the semi-labile biomolecule concentration trends double in summer. Interestingly, the grid fraction with a significant trend tends to be smaller in spring, with values not greater than 7 % (see

435 Fig. A3a, Fig. A4c). Lastly, $PL_{sw}$ decreasing trend also persists in the Nordic Seas; however, somewhat weaker and stronger





than in summer for the Barents and Greenland seas, respectively. In the same manner, Baffin Bay has a higher absolute upward trend in summer compared to spring. In contrast to the other biomolecules, a significant trend in the Canadian Archipelago is unexisting.

Overall, the geographical distribution of $PL_{sw}$ trend has similar characteristics to the NPP changes, specially in the inner Arctic and towards the sea ice edges (Fig. 6b, d and Fig. A3b, d). This close agreement is expected, as $PL_{sw}$ is a direct product of phytoplankton carbon exudation. Nevertheless, in Southern Norwegian and Barents seas during summer, south of the sea ice edge, $PL_{sw}$ showed a slightly positive or nearly absent trend that could be caused by depleted DIN. Under this condition, the carbon-overflow hypothesis (Engel et al., 2004, 2020) could explain the higher phytoplankton exudation rates. Similarly, for the semi-labile groups, this applies for multiple regions. However, the trend for the majority of the Arctic subregions predominantly increases, in contrast to the negative trend seen in NPP and $PL_{sw}$. The discrepancies are explained by the formation of TEP, which shows closer patterns to the NPP, as they rapidly form after $PCHO_{sw}$ exudation and represent a loss to the biomolecule. Interestingly, this process is more evident in sea ice free regions.

The FESOM-REcoM modelled NPP trends presented here have similar geographic patterns to the yearly changes discussed by Arrigo and van Dijken (2015) and Lewis et al. (2020) for most Arctic seas. A NPP increase in the inner Arctic waters, and only little variations or a slight decline in the Nordic seas and Arctic outflow regions has been reported in satellite-based analysis for the period 1990–2012 by Arrigo and van Dijken (2015). Moreover, Cherkasheva et al. (2025) also confirmed for the Greenland Sea that no significant NPP trend is observed for the 1998–2022 time series, consistent with the minimal changes we find in this region. However, some discrepancies are visible in the Barents and Chukchi Seas when comparing the results in Arrigo and van Dijken (2015) to those presented here. Besides the extended range of years we simulated in our study, one of the driving differences is the separation of seasons considered in the analysis. For instance, our simulations extend beyond the 2012 and for the late summer months (July-August-September), which is usually the time by which nutrients are at their lowest in Arctic waters (see DIN concentration in (Schourup-Kristensen et al., 2014)) potentially leading to the discrepancies seen in the Barents Sea compared to Arrigo and van Dijken (2015) and Lewis et al. (2020). Lastly, the trends calculated in the Chukchi sea might not be representative of the region, given the limited area in which the trends are significant.

As stated in (Leon-Marcos et al., 2024), note that the computation of the biomolecules does not consider ocean temperature effects on phytoplankton exudation (Zlotnik and Dubinsky, 1989; Guo et al., 2022). Nevertheless, a mesocosms study by Engel et al. (2011) demonstrated that for polar waters, an increase in seawater temperature (from 0 to 6 $^{o}C$) leads to a faster production and larger accumulation of dissolved combined carbohydrates (analogous to $PCHO_{sw}$) with no impact on the dissolved amino acids (proxy for $DCAA_{sw}$). This could be relevant in the current Arctic warming conditions with SST anomalies of several Celsius degrees in summer (Steele et al., 2008) that continue to exist in future Arctic projections.

## 4.2 Historical and present trends in the PMOA emissions

Here, the pan-Arctic trends in sea ice extent and PMOA emission anomalies are investigated. Figure 8 shows the time series of averaged summer sea ice area and total PMOA emission anomalies with respect to the period mean for 1990–2019 simulated

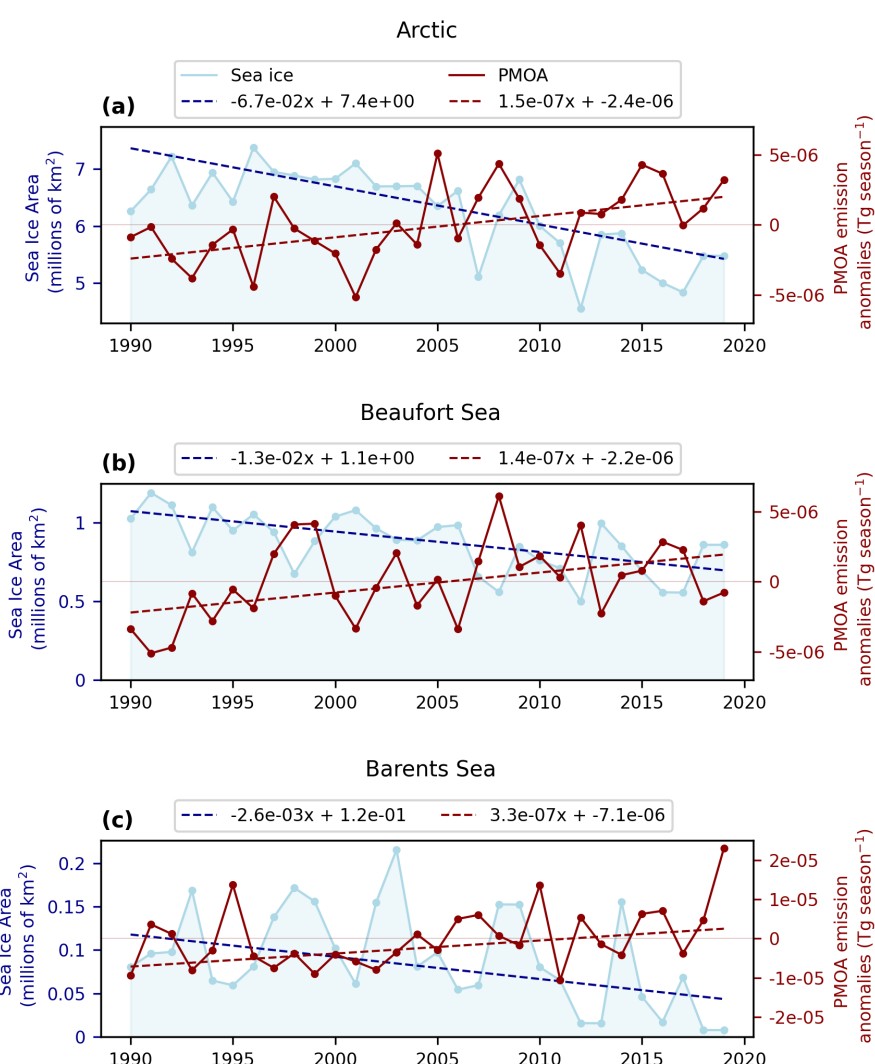

**Figure 8.** Time series of sea ice area in blue and averaged PMOA (PCHO$_{aer}$+DCAA$_{aer}$+PL$_{aer}$) emission mass flux anomalies in red for (a) the Arctic and (b) Kara Sea and (c) Barents Sea as defined in Fig. 1 for July-August-September of the simulated period 1990–2019 by ECHAM-HAM model. Dashed lines depict the trend line calculated using the slope and intercept values derived from the Theil–Sen slope estimator.



by the ECHAM-HAM model. The yearly mean values for the Arctic Ocean and preferred subregions within the Arctic circle are considered. Among all subregions, the ones presented here are the only cases for which both, sea ice area and POMA anomalies presented a significant trend over the 30-year period. Sen's slope value and intercept are always included. In addition, to have a better overview of the changes of the absolute aerosol quantities, the 15-year averaged values of PMOA flux and concentration along with sea ice area are shown in Table 2.

The sea ice area for the Arctic Ocean has suffered a critical decline after 2005 (Fig. 8a). A decreasing trend is visible throughout the period. This behaviour is obvious when comparing the extreme values. The maximum summer sea ice extent occurred in the first half of the period in 1996 with 7.4 million of $km^2$ in contrast to a minimum of nearly half 4.6 million of $km^2$ in 2012. Conversely, PMOA flux anomalies show an opposite trend to sea ice changes (Fig. 8a). Note that after 2005, positive anomalies are more frequent and stronger. Values were as low as $-5.1 \times 10^{-6}$ Tg season$^{-1}$ in 2001 and went up to $5.1 \times 10^{-6}$ Tg season$^{-1}$

in 2005. In contrast to the minimum sea ice area, the peak in the positive anomalies is reached earlier. Moreover, the changes of both variables between periods are not proportional. In addition to sea ice, other emission drivers as well as the ocean biomolecule abundance influence PMOA emissions, which vary for subregions within the Arctic. Hence, a moderated response of the fluxes to the sea ice retreat is evident in Fig. 8a. To illustrate the strong spatial variability and regional heterogeneities in the Arctic Ocean, the time series of Beaufort Sea and Barents Sea are discussed as examples.

The decline in sea ice area and the increase in marine emission anomalies are especially pronounced in the last decade of the study period (see Fig. 8b, c). In the Beaufort and Barents seas, positive marine emission anomalies occur more frequently during the second half of the period than in 1990–2004. Although the decline in Arctic sea ice area is stronger than in individual subregions, trends in marine emission anomalies remain similar across all cases. Figure 8b, c illustrates the intrinsic link between the fraction of open ocean and marine emissions. In most years, larger sea ice area corresponds to lower marine

aerosol anomalies, whereas smaller ice cover corresponds to higher fluxes.

For the Beaufort Sea, the magnitude of the emission anomalies is comparable with the Arctic mean (Fig. 8b). The largest positive and negative anomalies occurred in 2008 ($6.1 \times 10^{-6}$ Tg yr$^{-1}$) and 1991 ($-5.1 \times 10^{-6}$ Tg yr$^{-1}$). On the other hand, anomalies are stronger for the Barents Sea, given the larger fraction of open ocean (Fig. 8c). A prominent peak is seen in the last year of the studied period, with a value of $2.3 \times 10^{-5}$ Tg yr$^{-1}$. For this region, sea ice cover has a weaker effect on marine

aerosol occurrence.

To analyse the changes in other aerosol quantities, Table 2 summarizes the average emission fluxes and concentration in addition to sea ice over both halves of the simulated period. With this, we revealed the correlation between sea ice retreat and marine aerosol quantities. An increment of 17.3 % was attributed to the average Arctic PMOA emissions from 1990–2004 to 2005-2019, in contrast to a 16.5 % reduction in summer sea ice area. Similarly, PMOA concentration also grew by 7.7 %. The

rate of mean sea ice reduction in the Barents Sea from the early to the late fifteen years is the most notable. The decline is twice larger than that in the Beaufort Sea, with about 22 and 42 % decrease, respectively. The latter presents the most drastic increment in the emissions and aerosol concentration, rising more than 30 % and 40 %, respectively. However, fluxes in the Barents Sea experienced slightly more than half the increase detected in the inner Arctic sea, while aerosol concentration only rose by 4.5 %.



**Table 2.** Average sea ice area, PMOA concentration and accumulated emission mass flux over the Arctic and Arctic subregions Beaufort Sea and Barents Sea analysed in Fig. 8 for 15-year periods, 1990–2004 (I) and 2005–2019 (II) for July-August-September.

| | Arctic | | Beaufort Sea | | Barents Sea | |
|---|---|---|---|---|---|---|
| | I | II | I | II | I | II |
| Sea Ice area (km$^2$) | $6.8 \times 10^6$ | $5.7 \times 10^6$ | $9.7 \times 10^{-5}$ | $7.6 \times 10^{-5}$ | $1.2 \times 10^{-5}$ | $6.6 \times 10^{-4}$ |
| | $(3 \times 10^{-5})$ | $(6.4 \times 10^{-5})$ | $(1.3 \times 10^{-5})$ | $(1.6 \times 10^{-5})$ | $(4.7 \times 10^{-4})$ | $(5.1 \times 10^{-4})$ |
| PMOA flux (Tg season$^{-1}$) | $3.2 \times 10^{-2}$ | $3.8 \times 10^{-2}$ | $1.0 \times 10^{-3}$ | $1.4 \times 10^{-3}$ | $6.7 \times 10^{-3}$ | $7.8 \times 10^{-3}$ |
| | $(3.1 \times 10^{-3})$ | $(4.1 \times 10^{-3})$ | $(5.0 \times 10^{-4})$ | $(4.4 \times 10^{-4})$ | $(1.2 \times 10^{-3})$ | $(1.4 \times 10^{-3})$ |
| PMOA concentration (ng m$^{-3}$) | $2.0 \times 10^1$ | $2.2 \times 10^1$ | $8.0 \times 10^0$ | $1.1 \times 10^1$ | $4.3 \times 10^1$ | $4.5 \times 10^1$ |
| | $(1.8 \times 10^0)$ | $(2.1 \times 10^0)$ | $(3.8 \times 10^0)$ | $(4.1 \times 10^0)$ | $(5.8 \times 10^0)$ | $(7.8 \times 10^0)$ |

In spring, seasonal mean aerosol emission fluxes and PMOA concentrations across the Arctic are lower than in summer (Table A2), while sea ice cover is clearly broader. Although the decline in spring sea ice area is weaker than in summer, it remains detectable. Consequently, increases in aerosol emission fluxes during spring are less pronounced than in the warm season. PMOA concentration tends to decline in the second half of the modelled period. On the other hand, in the Beaufort Sea, the PMOA concentration increase during spring is less pronounced than that of summer. This might be related to the steep sea ice

loss in summer, with over 20 % reduction in the late fifteen years compared to only 3.1 % negative change through April-May-June. Lastly, for the Barents Sea, the variation in aerosol quantities are stronger for the early melting season despite the less variable sea ice area but slightly stronger SS emissions change rate.

### 4.3 Regional changes in PMOA emissions and budget

As the analysis shows, there is no uniform pan-Arctic trend in the emissions and occurrence of PMOA. Figure 9 illustrates the sea ice concentration in ECHAM-HAM simulations (from AMIP) and the regional trends of SIC and $\text{PL}_{aer}$, $\text{PCHO}_{aer}$ and SS emission flux across the Arctic as computed with ECHAM-HAM. The changes per unit of SIC of $\text{PL}_{aer}$ emission mass fluxes are also presented. Additionally, Figure 10 shows the trends of marine aerosol fluxes per region within the Arctic circle. Due to the high variability of surface winds, the 10m-Wind velocity trend has overall low significance in the Arctic (see Fig. A5)

and therefore not included in Figure 10.

The strongest sea ice concentration variations occurred at the outer edges of the ice pack (for SIC < 80 % in Fig. 9a). A significant loss in sea ice is evident for most areas in the Arctic (Fig. 9b). The strongest decrease occurs in the Chukchi and Beaufort Seas (see Fig. 10a). Nonetheless, for all regions, a decline of SIC predominates. Nonetheless, a few regions, such as the northern Canadian Archipelago and the north coast of Greenland, exhibit areas with a slight, statistically significant positive

trend.

As melting sea ice uncovers ocean areas where bubble bursting process may occur, the aerosol emission fluxes increase in the Arctic due to larger areas of open ocean water fraction (Fig. 9c–e). The strongest changes in PMOA and SS emission mass



fluxes are seen in the Southern Barents Sea and in the Greenland and Norwegian Seas (see Fig. 9c). SST and surface wind speed are also determinant in the emission fluxes estimation. These drivers led to strong emissions over these seas, which are mostly

ice free (Fig. A5). In contrast, a decrease over some areas of the North Atlantic waters is probably a result of weakening wind conditions. In the eastern Arctic, marine aerosol emissions are favoured by the reduction in SIC (Fig. 10a). Similar patterns over these regions are seen for $PL_{aer}$ and $PCHO_{aer}$ (see Fig. 9d, e and Fig. 10b–d). Note that overall, marine organic groups emissions trend's surface distribution align over the Arctic.

Some areas in the Chukchi, Kara and East Siberian Seas show a reduction in the marine emissions which mis more prominent

for PMOA species (Fig. 9d, e). For the last two cases, the changes could be associated with the slight increment in SIC (Fig. 9b). Furthermore, 10m-Wind variations generally occur in contrast to the SIC distribution (see Fig. A5a), weakening over zones of larger SIC due to a higher surface roughness.

The inverse relationship of emission fluxes and SIC is illustrated as well in the changes of emission mass fluxes per unit of SIC (Fig. 9f). Given the proportional dependency of the emissions on the open ocean fraction per grid cell, a negative correlation

was expected. Over the Arctic, changes of $PL_{aer}$ with respect to SIC are as low as -0.7 ng m$^{-3}$ per unit of SIC. The strongest negative correlation is found towards the ice edges for the marine biomolecules. For regions with sea ice concentrations under 20 % subject to drastic modifications throughout the season and years, the changes of emission per unit of SIC were strongly negative, and we excluded them from the analysis.

The average estimated increase for marine aerosols is shown in Fig. 10b–e. Note that for some regions and species, the trends

of the average regional emissions were not significant (blank spaces in Fig. 10). Among the Arctic subregions, the Greenland, and Norwegian, Barents, and Beaufort Seas are the only areas in which all sea spray components simultaneously increased. In contrast, for the Canadian Archipelago, Baffin Bay, Central Arctic, East Siberian Sea and Kara Sea, no significant trend for the 30-year period is detected. The strongest growth in flux occurred in the Greenland and Norwegian Seas for all marine species, followed by the Barents Sea (Fig. 10b–e). Similarly, for inner Arctic seas, fluxes rise considerably in agreement with

the strongest sea ice reduction (Fig. 10a). Note that changes are not statistically significant for $PL_{aer}$ in the Russian shelf.

In contrast to the summer months, the occurrence of emissions through April–June period, is limited to the Barents, Greenland and Norwegian Seas (Fig. A6). Whereas, weaker absolute changes are seen for SS in this period, the trend of the emission flux of PMOA species is stronger than in July-August-September. Surface patterns strongly diverge among marine species. $PCHO_{aer}$ flux (Fig. A6d) notably increases over the North Atlantic basin. For areas where SS (Fig. A6c) indicated a decrease,

the organic specie's trend is nearly absent, except off the coast of Norway. $PL_{aer}$ (Fig. A6d), on the other hand, presents a different and opposite distribution to $PCHO_{aer}$ in the Greenland Sea.

Since fluxes patterns are not identical among biomolecules, differing and even contrasting regional trends are seen. Equally, the diverse biomolecule abundance in the ocean, as well as their physico-chemical characteristics, explain why the fluxes trends are not aligned to that of SS in all cases. Some evident patterns could be seen in $PL_{aer}$ emission trend in the Chukchi Sea,

which coincides with the $PL_{sw}$ ocean concentration changes (Fig. 6b) with decreasing flux but not with SS emission. This emphasises the influence that the ocean biological activity has on marine aerosols and how variable are the emission per region within the Arctic Ocean.





In summary, SIC changes are relevant in the inner Arctic, controlling the areas where marine emissions can occur, altering SST and wind stress. On the contrary, in sea ice free ocean conditions, surface winds and SST sway the emission occurrence.

The comprehensive analysis of the marine biomolecules ocean concentration in comparison to the aerosol emission changes indicates that for most Arctic regions, the marine bioactivity also plays a critical role in the organic aerosols emissions.

Lastly, to analyse the relative changes per year of each marine species over the 30 years across Arctic subregions, Fig. 11 shows the percent of change per year of emission flux and aerosol concentration. For the whole Arctic, SS emissions increased by $1.2 \% \, \mathrm{yr}^{-1}$. Among PMOA aerosols, $\mathrm{PCHO}_{aer}$ present the strongest relative increase compared to $\mathrm{DCAA}_{aer}$ and $\mathrm{PL}_{aer}$.

For the Arctic subregions, despite the absolute values being the highest for the Barents, Greenland, and Norwegian waters (Fig. 10b–e), the relative increase is stronger for the inner Arctic seas. The Beaufort and Laptev seas have strong positive values, ranging between $2.1 \% \, \mathrm{yr}^{-1}$ and $3.2 \% \, \mathrm{yr}^{-1}$. Aerosol concentration trends, on the other hand, are only statistically significant for all species in the Beaufort Sea. Besides this region, SS is only relevant for the whole Arctic and Chukchi Sea, while $\mathrm{PCHO}_{aer}$ trends are additionally significant in the Canadian Archipelago and Laptev Sea. Note that, given the complex

transport and deposition processes that aerosols undergo once emitted, the trends of aerosol concentration does not necessarily reflect that of the emission fluxes. They are smaller in magnitude, spanning from 1.1 to up to $2.7 \% \, \mathrm{yr}^{-1}$ for the Arctic subregions. For the Arctic, quantities are slightly weaker than for the emissions and only an increase of 0.6 and $0.7 \% \, \mathrm{yr}^{-1}$ occurs for SS and $\mathrm{PCHO}_{aer}$, respectively. Note that $\mathrm{PCHO}_{aer}$ is generally the organic group with the most prominent augment across Arctic subregions. Conversely, for the early melting season (April-June), while statistically significant trends were

barely apparent for the aerosol concentrations, upward emission trends for some species are found in the Barents, Norwegian, Kara, Laptev and Chukchi seas. Values tend to decrease for the Canadian Archipelago and Baffin Bay (Fig. A7). Among all biomolecules, $\mathrm{PCHO}_{aer}$ is the only group with a trend for the whole Arctic, with a relative change exceeding that calculated in the summer.




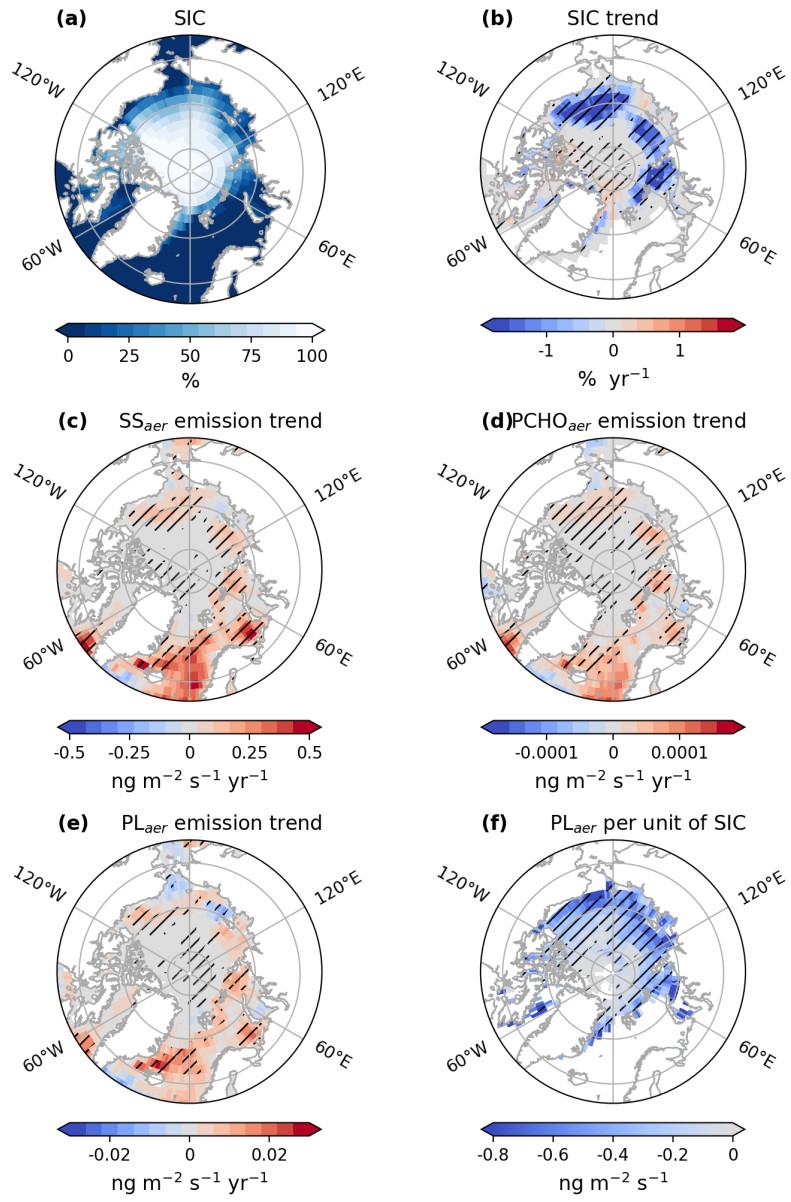

**Figure 9.** Maps of (a) average sea ice concentration (SIC), (b) trend of SIC, trends of emission fluxes of (c) SS, (d) $PCHO_{aer}$, (e) $PL_{aer}$ and (f) changes of emission fluxes of $PL_{aer}$ per unit of SIC for SIC>20 %, for July-August-September of the simulated period 1990-2019 by ECHAM-HAM model. The trend of $PL_{aer}$ per unit of sea ice was computed based on a linear regression model. The hatching indicates the areas over which trends are significant (Mann-Kendall test or t-test, p-value<0.05).





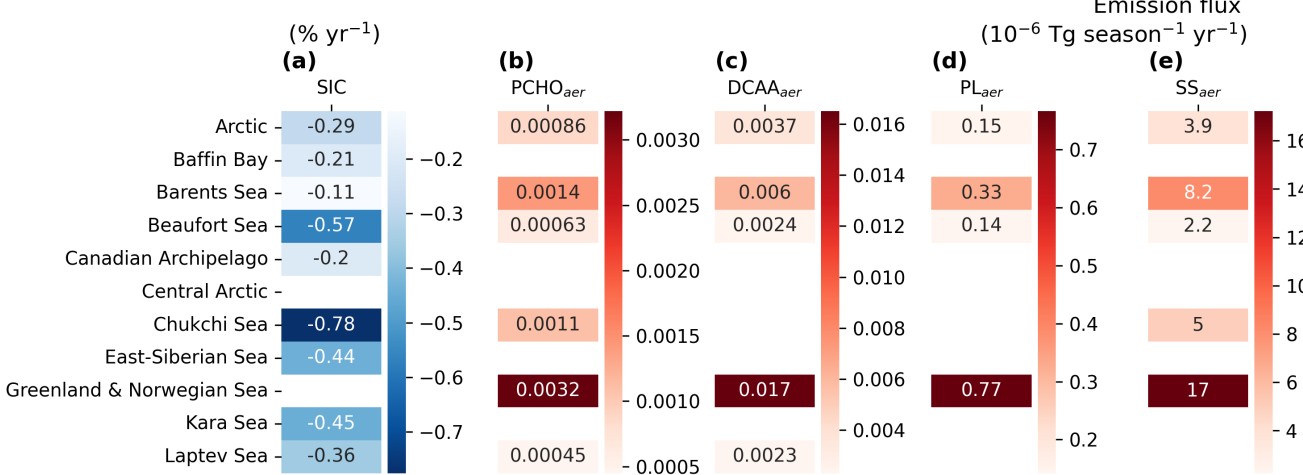

**Figure 10.** Heatmaps of trends averaged over the Arctic and subregions defined in Fig. 1 for (a) SIC, aerosol emission mass flux of (b) $PCHO_{aer}$, (c) $DCAA_{aer}$, (d) $PL_{aer}$ and (c) SS simulated by ECHAM-HAM model for July-August-September of the period 1990–2019. Only regions where the trend was significant are included (Mann-Kendall test, p-value<0.05).



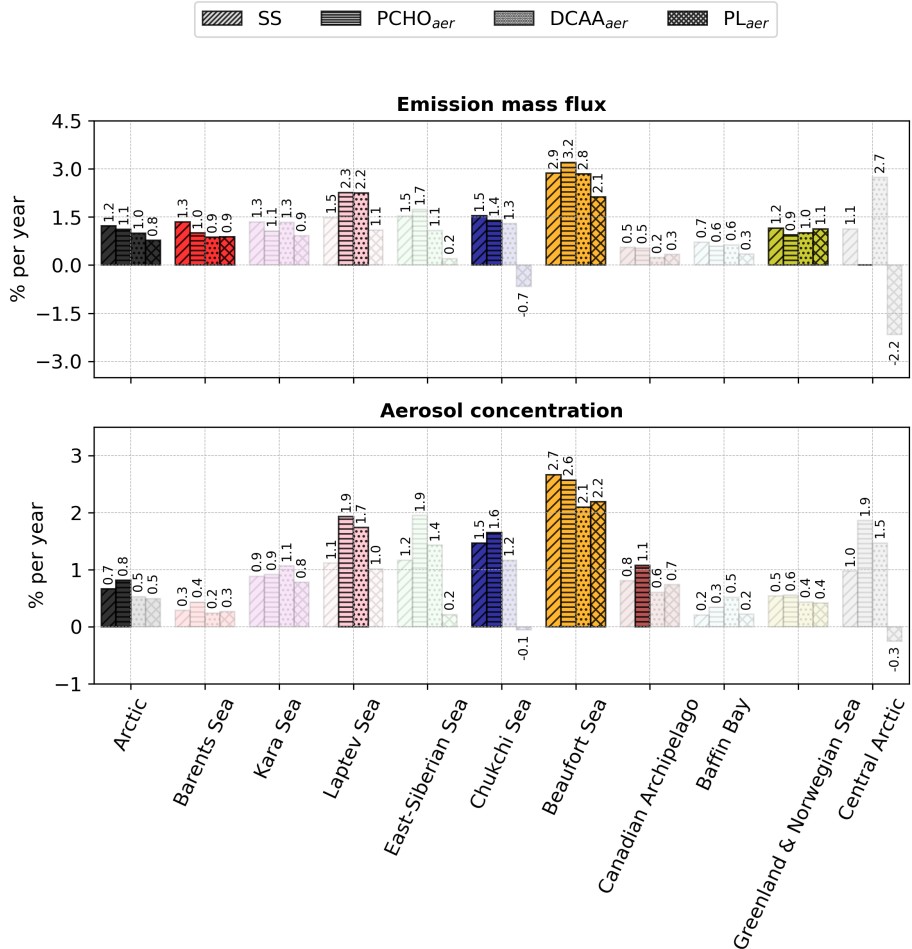

**Figure 11.** Bar plot of the percent of change per year of emission flux and near-surface aerosol concentration of marine species for the Arctic and subregions defined in Fig. 1 for July-August-September of the period 1990–2019. Values were calculated by normalizing the slope of the trend analysis by the 30-year average value for every subregion. The values atop the bars are the corresponding percent per year. The shaded bars represent the cases with not significant trend (Mann-Kendall test, p-value>0.05).





## 5    Challenges of modelling PMOA

Observational records are too brief and geographically scarce to establish robust trends. The limited availability of marine organics' seawater samples from Arctic field campaigns, and the lack of aerosol-species resolved observations, constrains further improvement of methods for computing ocean biomolecules and marine aerosol emissions in the polar region. This data scarcity is particularly evident in the species-resolved model outputs of the present study. What is presented here is therefore the best possible estimate of pan-Arctic and subregional conditions, given current data. Nevertheless, inherent uncertainties

must be taken into account when evaluating the results.

The climate-driven sea ice reduction with the subsequent appearance of wider open ocean areas contributes to an increase in marine emissions. Aerosol-climate model studies agree on a further increase in SS aerosol budget in the coming decades, with a relevant impact on cloud formation and cloud-radiative effects in the Arctic (Struthers et al., 2011; Gilgen et al., 2018; Lapere et al., 2023). Yet, large model uncertainties remain in the representation of marine organic aerosol sources and sea salt

emission (Lapere et al., 2023). Accounting for all relevant aerosol-related processes represents a major challenge for models in the Arctic (Schmale et al., 2021; Whaley et al., 2022), specially for large-scale models (Ma et al., 2014). Moreover, aerosol source apportion, mixing and removal mechanisms should be improved in models as they are the origin of significant uncertainties (Wang et al., 2013; Schmale et al., 2021; Whaley et al., 2022). Aerosol-cloud interaction and their impact on Arctic mixed-phase clouds remains highly uncertain, and considering them in models is difficult (Morrison et al., 2012). Furthermore,

the representation of other important marine aerosol sources besides open ocean could represent a limitation in most aerosol models. Recent findings by Lapere et al. (2024) raise the necessity of further researching the SS emission from leads, as their contribution could be comparable to the averaged open-ocean SS fluxes. As observations have linked organic aerosols and biological components in seawater samples from leads (May et al., 2016; Kirpes et al., 2019), neglecting this marine source in models could potentially underpredict the actual PMOA concentration over the ice pack.

Importantly, the source functions to account for marine emission are parameterized in various ways, essentially following the correlation between the surface wind speed and the sea spray fluxes (Mårtensson et al., 2003; Gong, 2003). Nevertheless, the performance of SS emission schemes in models varies over a wide range (Neumann et al., 2016; Barthel et al., 2019; Lapere et al., 2023). These differences gain relevance in the PMOA fluxes estimation, since the SS scheme and model configuration determine the emission patterns and PMOA budget (Leon-Marcos et al., 2024).

In the representation of marine biogenic emissions, some challenges rise in terms of PMOA components. Firstly, DOC sources in seawater encompass many other generation mechanisms than phytoplankton carbon exudation along (Carlson, 2002). Hence, ocean concentration of organic aerosol precursors could slightly diverge from our results, depending on the approach to modelling ocean organic groups (Burrows et al., 2014; Ogunro et al., 2015). Secondly, despite being integrated in the FESOM-REcoM model as a tracer, a parameterization to account for the aerosolization of TEP or their enrichment in aerosols has not

been developed and therefore, not considered here. To our knowledge, the implementation of marine gel-like particles has not been included in aerosol-climate models. Nevertheless, given the observational evidence of their contribution to marine Arctic aerosol and CCN (Leck et al., 2002; Leck and Bigg, 2005a; Orellana et al., 2011; Leck et al., 2013), it is a topic worth exploring



in future research. Lastly, other components we neglect are marine microorganisms and bacterial cells, which could also be transferred to aerosols through bubble bursting (Bigg and Leck, 2001; Fahlgren et al., 2015; Zinke et al., 2024), in addition
to the potential atmospheric biochemical activities of these airborne microorganisms (**?**Ervens and Amato, 2020; Zeppenfeld et al., 2021, 2023). Despite these shortcomings, the current study's results reflect the major trends based on the current state of knowledge.

## 6 Summary and Conclusions

As Arctic sea ice continues to melt, elucidating the response of marine organic aerosol emission is important, as they are a potentially important climate factor, particularly at high latitudes. In the current study, we investigated the distribution pattern and seasonality of three main marine biomolecule groups in the Arctic Ocean: dissolved carboxylic acidic containing polysaccharides (PCHO), dissolved combined amino acids (DCAA), and polar lipids (PL). These components are included within the model ECHAM-HAM as aerosol tracers to account for the emission, transport, and interactions with clouds and radiation.

The geographical distribution of biomolecule groups depends on the production and loss mechanisms considered in their computation. The physico-chemical characteristics of organics in seawater determines their transfer to aerosols. PL group is the most relevant to POMA and the occurrence in seawater concentrates mostly in coastal regions with river mouths, that provide nutrients to the Arctic seas. Seasonal patterns of the marine biomolecules and organic mass fraction in nascent aerosols have a remarkable seasonality. Maximum modelled contributions of the three organic groups typically occur between May and July.

The distributions of marine aerosols and their analogous in seawater strongly vary across Arctic subregions. The diversity is determined by riverine nutrients supply, sea ice conditions and ocean vertical mixing.

The PMOA emission fluxes were also analysed and tend to be stronger in North Atlantic waters during winter (January-February-March), spreading towards the central Arctic as sea ice melts in summer. Total annual average PMOA emission mass flux and atmospheric burden are $5.7 \times 10^{-2} \, \mathrm{Tg \, yr^{-1}}$ and $1.6 \times 10^{-4} \, \mathrm{Tg}$, respectively. Overall, aerosol quantities have risen
for 2005-2019 with respect to the preceding fifteen years. This increase across the Arctic varies by species group, influenced by regional dependencies, differences in bloom peak timings, and the efficiency of atmospheric aerosol wet removal.

As PMOA is emitted together with SS, its distribution matches in most cases that of SS fluxes. Nevertheless, the seasonality for Arctic subregions shows the critical influence of marine biological activity, causing a bimodal seasonal distribution in contrast to the unimodal Arctic average of SS emissions. PMOA fluxes initially peak in May, driven by the Greenland, Norwegian, and
Barents seas' contribution, decaying towards June with the minimum in SS fluxes. This is followed by a slightly higher maximum in September, concurring with the lowest SIC in the inner Arctic seas. We attribute the PMOA patterns to the influence of surface wind, open ocean fraction and biomolecule ocean concentration, and to a lesser degree to the SST variations.

The 30-yr historical Arctic trends demonstrates that the negative changes in sea ice concentration and changing primary production significantly impact phytoplankton exudation. Whereas, a rise in total marine biomolecule mass was detected in most
Arctic inner seas, a decreasing or contrasting trend occurs in the outflow regions. In terms of aerosols, summer (July-August-





September) emission flux anomalies have large interannual variations, with a general tendency to increase with declining sea ice for the second half of the studied period. As for the ocean, PMOA trends have noticeable differences among Arctic sub-regions, with predominantly positive changes. PMOA groups show a variable response. We found that the Arctic average emission fluxes of $PL_{aer}$, $DCAA_{aer}$ and $PCHO_{aer}$ have increased by $8.6 \times 10^{-4}$, $3.7 \times 10^{-3}$ and $1.5 \times 10^{-1}$ Tg season$^{-1}$ yr$^{-1}$

655 respectively, since 2019. This represents a relative change of 1.1, 1 and 0.8 % yr$^{-1}$ for each group.

The results of this modelling study indicate that PMOA emission are sensitive to the sea ice retreat and changes in marine primary productivity. The heterogeneous evolution of PMOA species spanning from 1990–2019 suggests that the individual components of PMOA could have different influences on cloud and precipitation formation. Our work provides a model setup, which accounts for different marine organic aerosols groups, that will be extended to consider other marine sources and aerosol-

660 cloud interaction processes in upcoming works. Considering distinct properties of cloud condensation and ice nucleation could have varying impacts on cloud formation and associated climate effects. In this study, we found that PCHO followed by DCAA held the most prominent relative changes in aerosol quantities for the Arctic circle and most subregions. Due to the enhanced ice-nucleating activity associated with these groups, we could speculate that their contribution to INP will also experience some increase, potentially leading to a positive cloud radiative effect.

665 *Code and data availability.* Interactive computing environments for data processing and figure generation can be found at https://doi.org/10.5281/zenodo.15582702. ECHAM-HAM model is made available to researchers under the HAMMOZ Software Licence Agreement, which outlines the usage conditions for the model (https://redmine.hammoz.ethz.ch/projects/hammoz/wiki/1_Licencing_conditions, last accessed: 22 November 2024). The version employed in this work, including the implementation for primary marine organic aerosol emissions, is archived on Zenodo (https://doi.org/10.5281/zenodo.14193491). The simulation setup files and code for integrating primary marine aerosols

670 into the model are provided at https://doi.org/10.5281/zenodo.14203456. The source code for the FESOM2.1-REcoM3 model is also publicly available at https://doi.org/10.5281/zenodo.14017536. Additionally, the biogeochemical model tracers used to derive marine biomolecule groups and ocean biomolecule concentrations are available at https://zenodo.org/records/15172565. Data post-processing and trend analyses were conducted with python (Python Software Foundation version 3.10.10), utilizing libraries such as pymannkendall, xarray, pandas, and cartopy, seaborn and matplotlib for handling and visualizing model outputs. Finally, Climate Data Operators (cdo) version 2.2.4 were used

675 to adapt bottom boundary condition datasets to the ECHAM–HAM grid and to compute Arctic accumulated emission fluxes and burdens.





**Table A1.** Index of abbreviations for the most significant aerosol and marine compounds studied here.

| General terms | |
|---|---|
| PCHO | Dissolved carboxylic acidic containing polysaccharides |
| DCAA | Dissolved combined amino acids |
| PL | Polar lipids |
| **Seawater** | |
| DOC | Dissolved organic carbon |
| $DOC_{phy\_ex}$ | DOC fraction exuded by phytoplankton |
| $PCHO_{sw}$ | PCHO in seawater |
| $DCAA_{sw}$ | DCAA in seawater |
| $PL_{sw}$ | PL in seawater |
| TEP | Transparent exopolymer particles |
| **Aerosol particles** | |
| PMOA | Primary marine organic aerosol |
| SS | Sea salt |
| $PCHO_{aer}$ | PCHO in aerosol particles |
| $DCAA_{aer}$ | DCAA in aerosol particles |
| $PL_{aer}$ | PL in aerosol particles |



**Table A2.** Average sea ice area, PMOA concentration and accumulated emission mass flux over the Arctic and Arctic subregions Beaufort Sea and Barents Sea analysed in Fig. 8 for 15-year periods, 1990–2004 (I) and 2005-2019 (II) for April-May-June.

|  | Arctic | | Beaufort Sea | | Barents Sea | |
|---|---|---|---|---|---|---|
|  | I | II | I | II | I | II |
| Sea Ice area ($km^2$) | $1.2 \times 10^7$ | $1.1 \times 10^7$ | $1.5 \times 10^6$ | $1.5 \times 10^6$ | $6.8 \times 10^5$ | $5.1 \times 10^5$ |
|  | $(2.8 \times 10^5)$ | $(4.9 \times 10^5)$ | $(6.1 \times 10^4)$ | $(6.1 \times 10^4)$ | $(1.7 \times 10^5)$ | $(9.1 \times 10^4)$ |
| PMOA flux (Tg $season^{-1}$) | $1.8 \times 10^{-2}$ | $1.9 \times 10^{-2}$ | $5.29 \times 10^{-5}$ | $6 \times 10^{-5}$ | $4.4 \times 10^{-3}$ | $5.4 \times 10^{-3}$ |
|  | $(4 \times 10^{-3})$ | $(4 \times 10^{-3})$ | $(8.3 \times 10^{-5})$ | $(4.8 \times 10^{-5})$ | $(1.4 \times 10^{-3})$ | $(1.3 \times 10^{-3})$ |
| PMOA concentration | $1.1 \times 10^1$ | $1.1 \times 10^1$ | $9.6 \times 10^{-1}$ | $1.1 \times 10^0$ | $2.6 \times 10^1$ | $3.0 \times 10^1$ |
| (ng $m^{-3}$) | $(1.9 \times 10^0)$ | $(2.7 \times 10^0)$ | $(8.5 \times 10^{-1})$ | $(6.2 \times 10^{-1})$ | $(8.2 \times 10^0)$ | $(1.1 \times 10^1)$ |



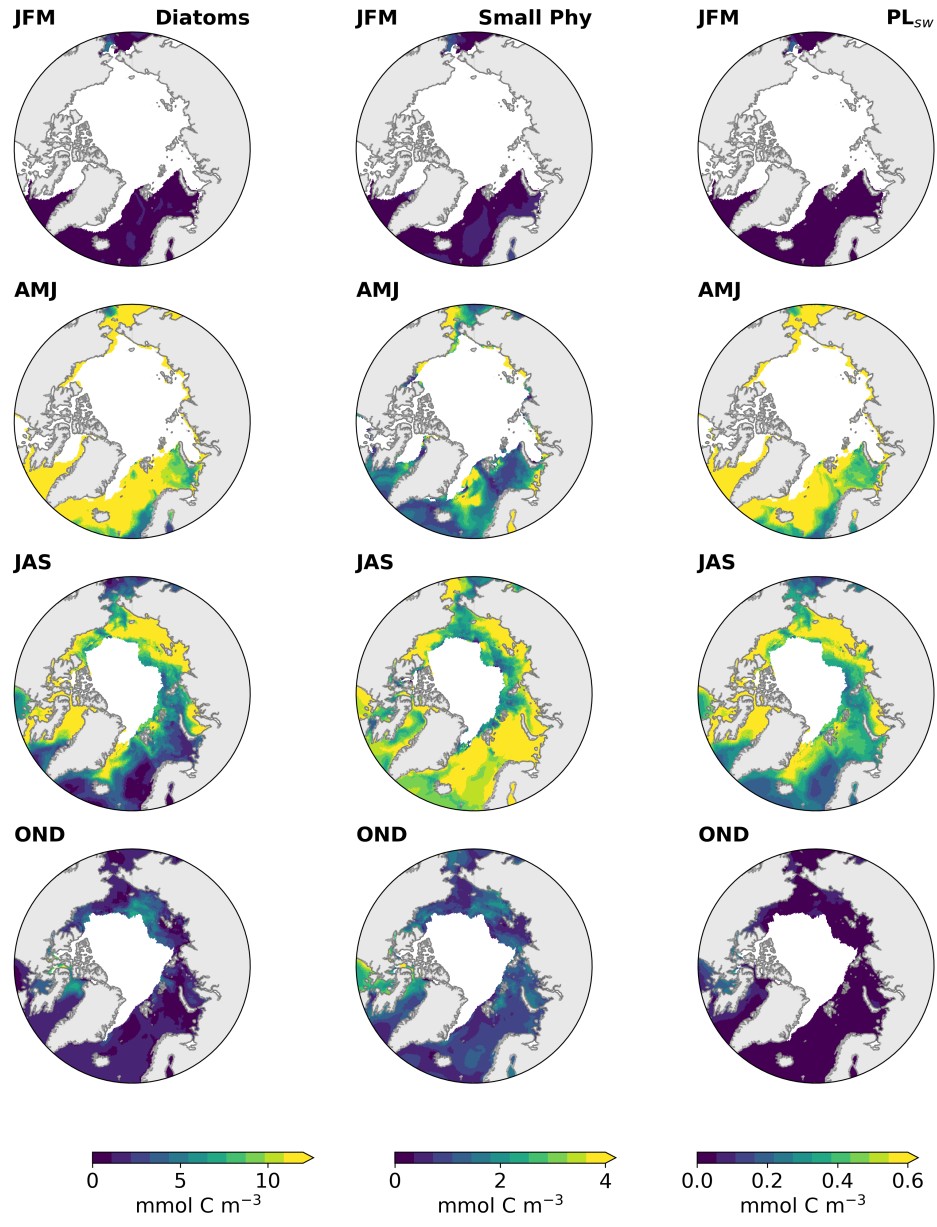

**Figure A1.** Maps of the carbon concentration of phytoplankton groups simulated by REcoM, Diatoms (left panel), small phytoplankton (middle panel) and $PL_{sw}$ for January-February-March (JFM), April-May-June (AMJ), July-August-September (JAS) and October-November-December (OND) for the period 1990-2019 and sea ice free ocean conditions (SIC<10 %).



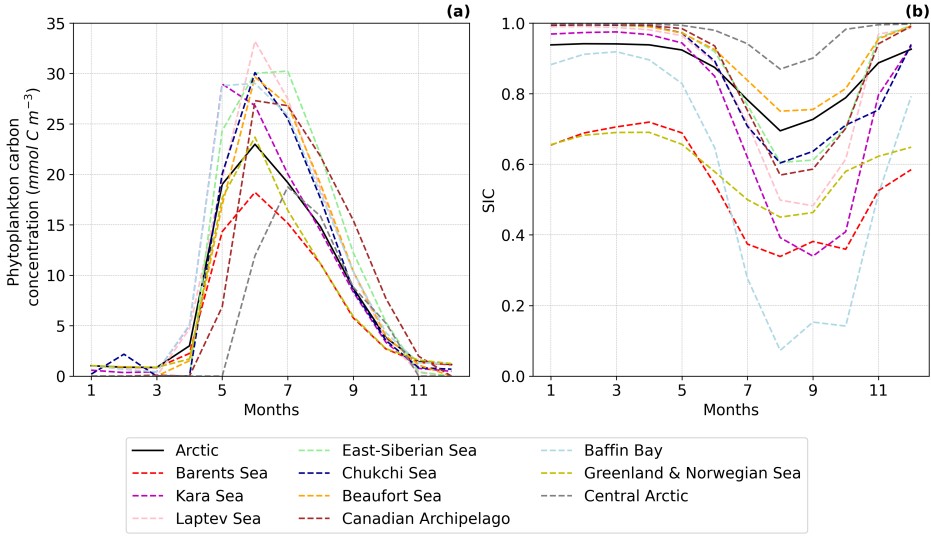

**Figure A2.** Seasonal climatology of (a) ocean phytoplankton carbon concentration with sea ice free ocean conditions (SIC<10 %) and (b) SIC modelled by FESOM for the period 1990-2019 averaged over the Arctic Ocean ($63°N − 90°N$) and Arctic subregions in Fig. 1.



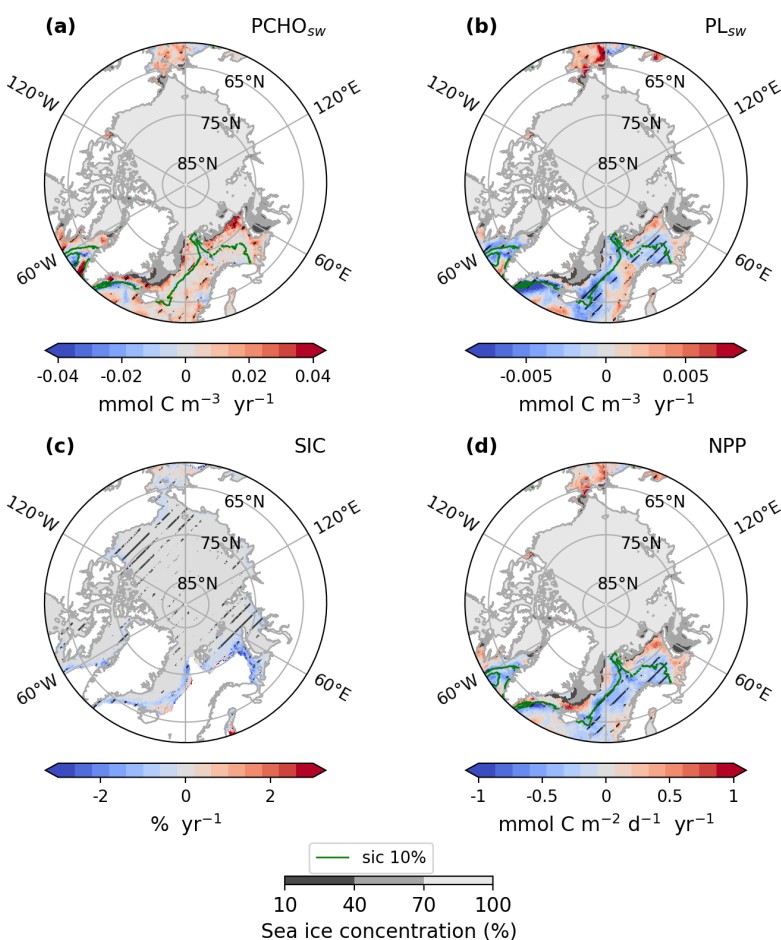

**Figure A3.** Arctic trends of (a) PCHO$_{sw}$ and (b) PL$_{sw}$ ocean concentration, (c) sea ice concentration and (d) net primary production from REcoM model for April-May-June of the simulated period 1990–2019. The hatching indicates the areas over which trends are significant (Mann-Kendall test, p-value<0.05). The green contour line depicts the average season 10 % sea ice concentration. The minimum seasonal SIC for the period occurred in June 2016, and it is also represented in shaded gray.



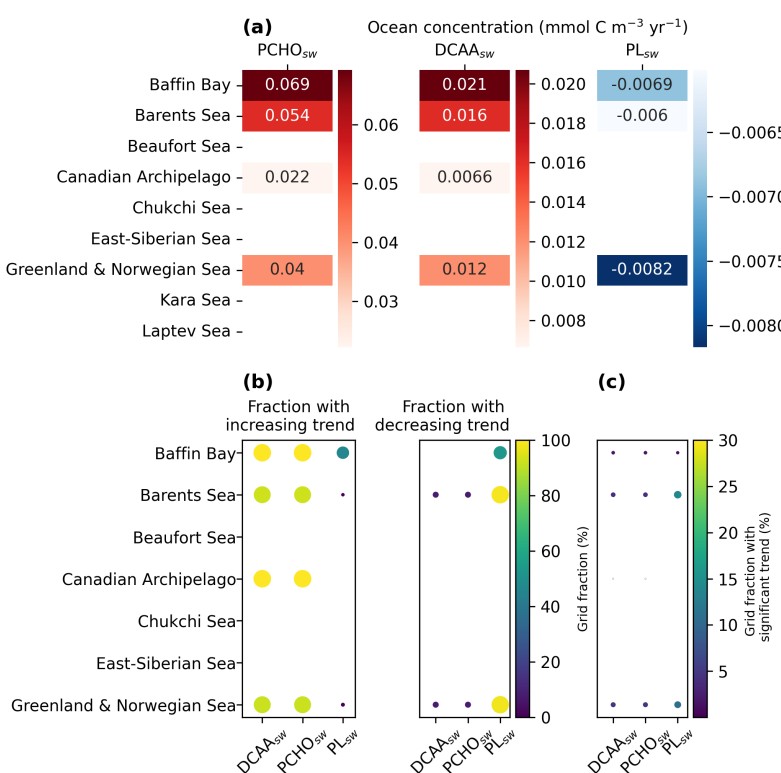

**Figure A4.** Maximum trends for regions in Fig 1 of the (a) biomolecule ocean concentration trend with the highest grid fraction of increasing or decreasing trends of (b) ocean concentration for April-May-June of the period 1990–2019. Only cases where the trends are significant (Mann-Kendall test, p-value<0.05) are considered, and (c) illustrate the fraction they represent of each region in terms of percent of grid cell. Values were obtained after applying a mask with the minimum sea ice concentration shown in Fig. A3.





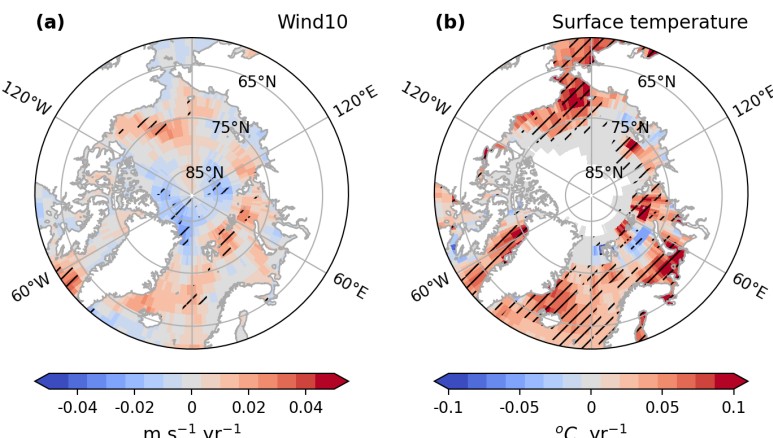

**Figure A5.** Arctic trends of (a) surface wind speed sea and (b) sea surface temperature (SST) for July-August-September of the simulated period 1990-2019 from ECHAM-HAM model. Only grid cells where the trends are significant (Mann-Kendall test, p-value<0.05) are considered.

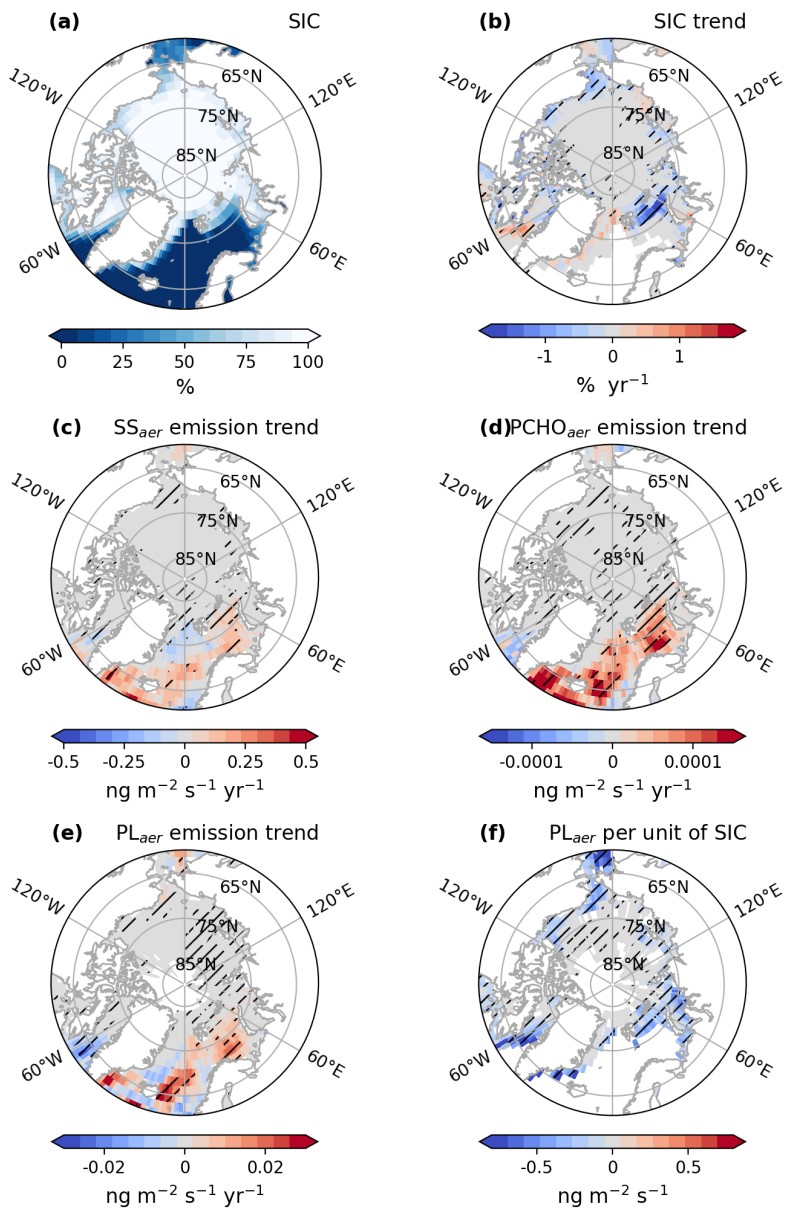

**Figure A6.** Maps of (a) average sea ice concentration (SIC), (b) trend of SIC, trends of emission fluxes of (c) SS, (d) $PCHO_{aer}$, (e) $PL_{aer}$ and (f) changes of emission fluxes of $PL_{aer}$ per unit of SIC for SIC>20 %, for April-May-June of the simulated period 1990-2019 by ECHAM-HAM model. The trend of $PL_{aer}$ per unit of sea ice was computed based on a linear regression model. The hatching indicates the areas over which trends are significant (Mann-Kendall test or t-test, p-value<0.05).

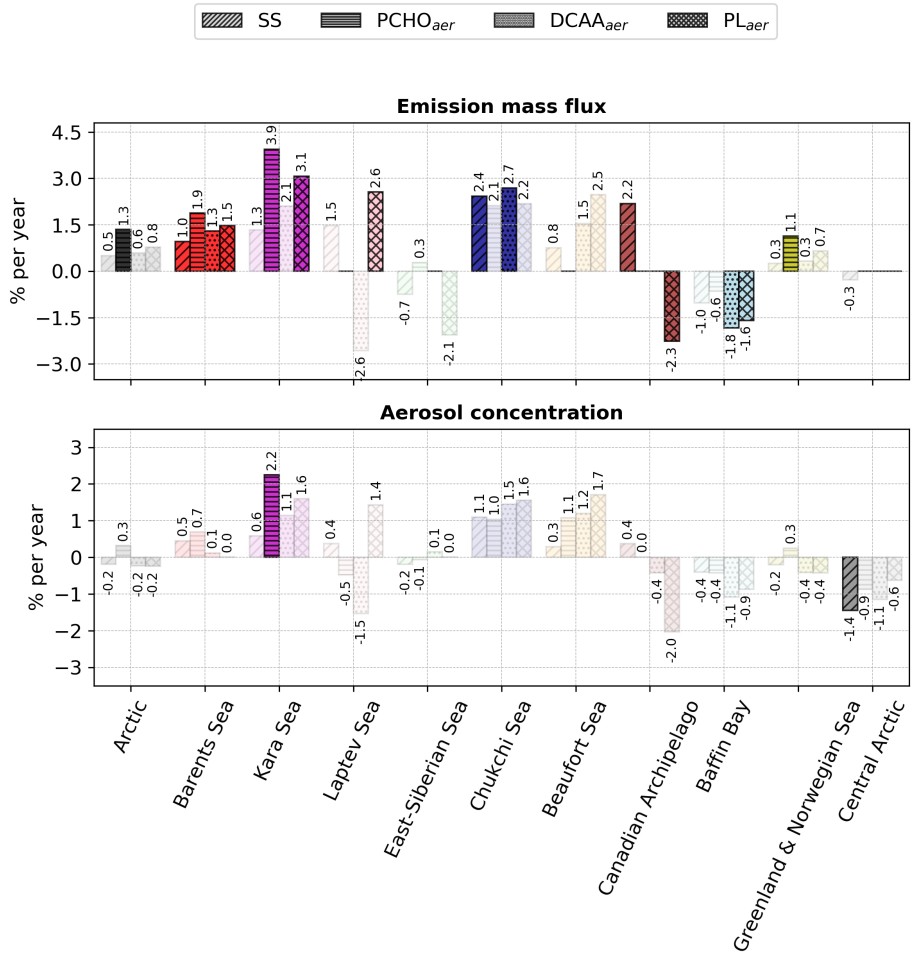

**Figure A7.** Bar plot of the percent of change per year of emission flux and aerosol concentration of marine species for the Arctic and subregions defined in Fig. 1 for April-May-June of the period 1990–2019. Values were calculated by normalizing the slope of the trend analysis by the 30-year average value for every subregion. The values atop the bars are the corresponding percent per year. The shaded bars represent the cases with not significant trend (Mann-Kendall test, p-value>0.05).



*Author contributions.* AL-M and BH designed the aerosol-climate model simulations. AL-M carried out the simulations, post-processed the output, and analysed marine biomolecules and aerosol tracers, as well as performed the trend analysis. MZ post-processed the biogeochemistry model output and contributed to discussions on Arctic marine biogeochemistry. BH and MZ provided feedback on technical aspects in computing the trends. MvP and SZ offered scientific guidance on organic processes in seawater and Arctic aerosols. AB streamlined and structured the initial stages of this work and gave insightful feedback on the trend analysis. LO helped enhance the scientific quality and completeness of the results. AL-M drafted the manuscript, which was subsequently reviewed and edited by MZ, MvP, SZ, AB, IT, LO, and BH. BH also provided supervisory guidance and scientific input throughout the manuscript. Funding was secured by AB, BH, and MvP.

*Competing interests.* The authors declare that they have no conflict of interest.

*Acknowledgements.* We gratefully acknowledge funding from the Deutsche Forschungsgemeinschaft (DFG; German Research Foundation; project number 268020496; TRR 172) under the Transregional Collaborative Research Centre "ArctiC Amplification: Climate Relevant Atmospheric and SurfaCe Processes, and Feedback Mechanisms (AC)[3]." We also thank the Deutsches Klimarechenzentrum (DKRZ) for the computing time provided under project number bb1005. We appreciate the computing time provided by the Resource Allocation Board on the Lise and Emmy machines at NHR@ZIB and NHR@Göttingen as part of the NHR infrastructure. The FESOM-REcoM model simulations were conducted with computing resources under the project hbk00084. We thank Dr. Susannah Burrows (Pacific Northwest National Laboratory) for her assistance in the offline setup of OCEANFILMS and for providing ocean macromolecule data from CESM biogeochemical modules for our test runs. We appreciate Dr. Michael Weger's (TROPOS) assistance in the implementation and optimization of some post-processing python tools employed in the trend analysis.



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
