# Peer review of "Thirty Years of Arctic Primary Marine Organic Aerosols: Patterns, Seasonal Dynamics, and Trends (1990–2019)"

_EGUsphere, 2025_

## Author Comment (AC2)

**Response to RC2: 'Comment on egusphere-2025-2829'**

**October 22, 2025**

We are grateful to the reviewer for taking the time to review our manuscript and for offering such valuable feedback. Your observations and helpful recommendations contributed to enhancing the clarity and quality of our work. Please find below the response (in blue) to your comments (in black)

1. Abstract: what you mean by accumulated aerosol burden? It has not been introduced before so reader might find it difficult to understand what you mean by accumulated We agree with the reviewer that the concept of accumulated burden has not been clearly introduced at this point, and its meaning may therefore be unclear. We now use the more common term, "total aerosol burden", which is calculated as follows: Annual values were obtained by aggregating daily results from all grid cells across the Arctic region, which were then averaged over the 30-year period. We changed this in all the instances in the text. This also includes the

header of Table 1 in Section 3.3.1.

2. Abstract: These two lines almost say the same thing. Why you need both? "These quantities peak from May to September, coinciding with the phytoplankton bloom and seasonal sea ice minimum" and "Summer trend analysis (June–August) reveals a strong reduction in sea ice that correlates with rising concentrations of organic groups in seawater in the inner Arctic."

We understand that the overlap of the time periods in these two sentences may have caused confusion and given the impression of duplicated information. However, the sentences indeed refer to distinct findings of the analysis. To improve clarity, we have revised the text accordingly. We have removed the sentence "These quantities peak from May to September, coinciding with [...]" and modified the one before: "Results indicate that the strong seasonality in biomolecule concentrations and PMOA emissions is driven by marine productivity and sea salt emissions, with the peak occurring from May to September."

In the case of the second sentence, we slightly modified it for clarity: "Summer (June–August) trend analysis over the 30 years reveals a pronounced reduction in sea ice that correlates with rising concentrations of organic groups in seawater in the inner Arctic".

- 3. Abstract: This line says the rate of increase has decreased from 7 to 4% in the second half of the study. "Accumulated aerosol emissions and burdens over the Arctic increased by at least 7% and 4%, respectively, between the first and second halves of the study period.".
  - But this line says emission have become more frequent in last 15 years. "Positive emission anomalies have become more frequent over the past 15 years"

I think it's contradicting as your trends say the emission rate has rather decreased from 7 to 4% We understand that the sentence, as is, lacks clarity. The message is that the accumulated aerosol emissions increased by at least 7% and the burden increased by 4% from the first half (1990-2004) to the second half (2005-2019) of the studied period. While revising the text and results from Table 1, we noticed that the percentage value of the increase in emissions between periods (7%) was incorrect. Considering also this, we have modified the sentence in the Abstract as follows: "Total aerosol emissions and burdens over the Arctic increased by at least 12% and 4%, respectively, between 1990-2004 and 2005-2019."

Related to this topic, we added new results of the relative changes of the aerosol burden for all marine species. This can be found in section 4.3 and Fig. 12.

- 1. Discuss why the rate of increase has decreased from 7% to 4% Please, refer to the response to the previous comment.
- 2. Abstract: "PCHO showing the largest relative increase" mention a number here

  Thanks for the suggestion. We have included the amounts in the Abstract as follows: "... PCHO showing the largest relative increase, with 1.3 % and 0.8 % per year for the emissions and aerosol concentration, respectively."
- 3. Is the 'peak' season constant in a particular month for all the years? Mention this in the abstract if its changing or staying constant. Pernov (2022) mentioned peak MSA concentration will shift its peak month in next 50 years. It would be interesting to see if this shift in peak in other parameters has started showing in long-term data already

We thank the reviewer for this excellent question and valuable suggestion. We performed the analysis of the annual seasonality of marine biomolecules and PMOA emissions. For the emissions, the changes in the peak season are weak and not robust enough to support definitive

conclusions. However, ocean biomolecules did show indications of a shift in emissions towards an earlier month. This was included in the manuscript in lines 280-286 and lines 398-400:

"Lastly, we analyse the yearly seasonality in Arctic subregions to examine how the initiation and duration of biomolecule production have changed over the 30-year period. While the seasonal patterns remained stable for the Canadian Archipelago, Baffin Bay and, Barents, Greenland and Norwegian Seas, a pronounced interannual variability occurs for the inner Arctic seas. Among these, the Beaufort and Kara seas show strong indications that biomolecule release initiates one month earlier during the second half of the study period compared to 1990-2004 (see Fig. C1). Other studies based on satellite products have found trends in phytoplankton blooms shifting towards an earlier maxima (Kahru et al., 2011; Zhao et al., 2022). Similarly, recent modelling analysis by Manizza et al. (2023) also points towards earlier spring blooms in the inner Arctic seas."

"Analysis of the annual seasonality of PMOA emissions did not reveal a clear shift toward earlier onset. In the Beaufort Sea, emissions show a tendency to occur approximately one month earlier during the second half of the study period; however, the patterns are weak and not sufficiently robust to draw conclusions (not shown)."

Please find them in the text towards the end of sections 3.2 and 3.3.2, respectively. Fig. C1, Appendix C, was consequently added to the manuscript.

- 4. Line 43: "and the relevance of PMOA for cloud formation in the Arctic": add citations

  Thank you for the suggestion. We added the following citations: (Leck and Bigg, 2005; Bigg and
  Leck, 2001; Irish et al., 2017; Hartmann et al., 2021; Creamean et al., 2022; Porter et al., 2022).
- 5. Line 63: Cite Russell for carbohydrates. Also mention some literature on lipids.

  Thank you for the suggestion. We added Russell et al. (2010) reference in line 60: "... melt ponds (Zeppenfeld et al., 2023). This supports previous findings by Russell et al. (2010) of saccharide compounds in Arctic marine aerosols." We also included an additional reference related to this topic and to lipid's detection in line 63: "In addition to carbohydrate-like substances, Hawkins and Russell (2010), also found evidence of marine proteinaceous material in aerosol particles. Lipid-like molecules (e.g. n-alkanes and fatty acids) have also been analysed in the Bering Sea, with significant contributions to marine aerosols in summer Hu et al. (2023)."
- 6. Line 88: Any particular reason why only polysaccharides, amino acids and lipids were chosen

as components to represent PMOA? How much fraction of PMOA does each represent? "highly abundant" mention number here if possible

Note that the trend analysis of the organic species presented here is a follow-up study of Leon-Marcos et al. (2025), who introduced and thoroughly evaluated the species considered in this study. To clarify this, we added this reference to line 88.

According to observations, polysaccharides, amino acids, and lipids are abundant on the ocean surface and effectively transported to aerosols (Triesch et al., 2021a,b; Zeppenfeld et al., 2021; van Pinxteren et al., 2023; Jayarathne et al., 2023). This has also been confirmed in bubble bursting experiments by Rastelli et al. (2017). In the present Arctic-focused study and that by Leon-Marcos et al. (2025), the computed organic species represent a fraction of total PMOA. However, since the contribution and origin of each compound emitted via bubble bursting could significantly vary per region and season (e.g. Keene et al., 2007; Facchini et al., 2008; Russell et al., 2010), and measurement techniques and analysed compounds are not uniform (Cavalli et al., 2004; Facchini et al., 2008; Jayarathne et al., 2016; van Pinxteren et al., 2023), estimating what fraction of the total PMOA, the simulated groups in our study represent is not possible. Nonetheless, given the detailed evaluation against seawater and aerosol measurements performed by Leon-Marcos et al. (2025), the individual contribution of the simulated species is reasonably well represented.

7. Line 113: add citations about the accuracy of HAM model with the given assumptions. Do organics have large uncertainty in representation? If so, then taking results from this model and using them as input for another model will only lead to uncertainties. Its important to quantify this or at least cite.

A comprehensive evaluation of aerosol species within the standard ECHAM6.3–HAM2.3 model version was performed by Tegen et al. (2019). In addition, the marine organic species considered in the present studies were also thoroughly evaluated by Leon-Marcos et al. (2025) with species-resolved aerosol observations. As the model uncertainties are quantified in Leon-Marcos et al. (2025) and Tegen et al. (2019), additional evaluation would be beyond the scope and represent an overload for the scope of this paper.

Having said this, there may be a misunderstanding regarding the use of HAM (i.e., ECHAM6.3–HAM2.3) model results in another model. We think that the reviewer refers to the accuracy of the marine biogeochemical model FESOM-REcoM (e.g. Zeising et al., 2025), on which the computation of the biomolecule ocean concentration is based and serves as a surface boundary condition to the aerosol-climate model. The computed ocean biomolecules were also evaluated against seawater

samples by Leon-Marcos et al. (2025).

We agree with the reviewer that these points should be specified in the manuscript, and have included the corresponding references to the model evaluation studies as follows:

Lines 112-114 as follows: "The model includes several aerosol species such as sulphate (SO4), organic carbon (OC), black carbon (BC), mineral dust (DU) and sea salt (SS), which were evaluated by Tegen et al. (2019). Leon-Marcos et al. (2025) implemented PMOA species in the model as additional tracers in the accumulation size mode and performed a thorough evaluation of the model results."

Lines 134-136: "A more extensive explanation of the model characteristics, the methodology employed to compute the biomolecules, and the evaluation against seawater samples can be found in Leon-Marcos et al. (2025)."

**8. Line 6322; is it PMOA? Or POMA?**

We thank the reviewer for pointing that out. We have corrected and used the correct abbreviation "PMOA".

9. Line 655: "This represents a relative change of 1.1, 1 and 0.8 % yr-1 for each group." The group order mentioned is as follows: PLaer, DCAAaer and PCHOaer. So PCHO relative change is 0.8 % which is less then PL and DCAA. Then why In line 662, it is mentioned that PCHO has the most prominent relative change?

Thanks to the reviewer's comment, we noticed there was a mistake in that sentence, and the order of the percentages was incorrect. The correct order is 0.8, 1.1, 1.3  $\% \, \text{yr}^{-1}$  for  $\text{PL}_{aer}$ ,  $\text{DCAA}_{aer}$  and  $\text{PCHO}_{aer}$ , respectively. Hence,  $\text{PCHO}_{aer}$  has the most prominent change, as shown in the top panel of Fig. 11. This was corrected in the manuscript. Note that the percentages are different as they represent total emission fluxes (see also response to comment 2 of RC1).

10. It would be nice to include how much % change in CCN or INP do you think these relative changes in PMOA will cause. I ask this because the relative increase is around 1% which seems pretty small. So I want to get an idea if this small change will affect CCN/INP significantly. We agree with the reviewer that including the relative change in cloud parameters over the 30 years as a response to the PMOA trends would be very interesting. While this could indeed provide further scientific insight, it lies beyond the scope of the present manuscript and would require considerable additional effort in both model development and analysis. The paper, in its

present form, provides a consistent and coherent focus on aerosols, specifically, the climate-driven trends of PMOA species in the Arctic. Therefore, we plan to address this topic in future follow-up studies aimed at modelling and evaluating the role of PMOA-species-resolved interactions in Arctic clouds (i.e., CCN and INP activation) and their associated climate effects.

In the ECHAM6.3-HAM2.3 model, CCN is calculated for each aerosol mode following Abdul-Razzak and Ghan (2000) and encompasses the contribution of all species within a mode. However, the ice nucleation potential of PMOA has not been implemented yet in ECHAM6.3-HAM2.3. The latter is especially relevant in the Arctic, given the observational evidence of the dominance of biological INPs during summer (e.g. Irish et al., 2017; Hartmann et al., 2021; Porter et al., 2022; Creamean et al., 2022). Studies have also shown that some organic species (e.g. polysaccharides and proteins) could preferably activate as INP (McCluskey et al., 2018; Alpert et al., 2022). Hence, when examining the ice nucleation potential of PMOA, the marine aerosol composition should also be carefully considered. For the time available for answering the reviewers' questions, it is not feasible to provide a detailed analysis of aerosol-cloud and cloud microphysical processes.

Nevertheless, we performed an offline analysis as a test study to compute the nucleation potential of PCHO, the aerosol species undergoing the most significant relative changes. For the calculation, we used the novel parameterisation introduced by Hartmann et al. (2025) to estimate the INP concentrations from polysaccharides. Based on the PCHO concentration at the lowest model layer, we computed the INP concentration at -15 °C. We found reasonable results with climatological INP concentrations between 10-6 and 10-3 m-3 during the Arctic summer. In addition, we performed the trend analysis over the Arctic for 1990–2019 and found statistically significant increasing trends of nearly 1% yr-1, and higher rates for the inner Arctic subregions. Note that the relative changes are always provided in units of per cent per year, so a rate of 1% yr-1 represents a substantial change. Although this analysis sparked our interest to provide the scientific community with preliminary findings on the potential impact of marine species on INP concentrations in the Arctic, the results require further validation to be conclusive and well-founded.

11. Line 585. Agreed but it would be nice if you could also add these observational data points wherever and whenever available with a different symbol in your figures. That way we get an idea of how accurate your model estimates are.

The idea of this paragraph is to stress that there is no available data (i.e. seawater samples and aerosol observations) to derive trends in the Arctic. Nonetheless, for short-term observations, Leon-Marcos et al. (2025) performed a detailed evaluation of the model outputs of marine and aerosol variables against available species-resolved measurements, including those within the Arctic Circle (e.g. Svalbard and the Norwegian Sea). Following Leon-Marcos et al. (2025), we consider the results used for the trend analysis in this manuscript to be representative of regions without any observations available and expect them to behave similarly over time.

- 12. Line 620: typo: remove '?'
  - Thanks for pointing this out. This interrogation symbol was in place of a missing citation, which we have now incorporated.
- 13. Figure 1: add different observation stations in the Arctic (for reader to get a clearer picture)

  We thank the reviewer's suggestion; however, it is out of the scope of this manuscript to include
  a comparison of model results to observations, as this was previously performed by Leon-Marcos
  et al. (2025). Please refer to our response to comment 7, which specifies the reference to the
  preceding paper, which includes the model evaluation. For additional information, see also our
  response to comment 11.
- 14. Figure 3b. Maybe add residual as well to indicate how much fraction of total OMF is contributed by these three? Try including a stacked bar plot with errorbars for each month? Its nice when all components of OMF sums to 1.

We agree that representing the organic mass fraction (OMF) so that the sum of all contributions equals 1 would provide more complete information on the origin of each organic component. However, we believe there might be a misunderstanding regarding the concept of the OMF in the PMOA emission calculations. As defined in Section 2.1, the OMF refers to the species-specific fraction within the sea spray aerosol, not to the fraction of a species within the total organic component. Since the remaining organic fraction is not known from measurements, the focus here is limited to the contribution of the three species considered in this study (and in the preceding work by Leon-Marcos et al., 2025a). As discussed in Section 5, several components emitted via bubble bursting and contributing to PMOA are not considered in the study, and they would contribute to the remaining OMF.

The stacked plot is an excellent idea; however, this kind of plot is not suitable for this case as the species have different orders of magnitude, which we highlighted by adding two y-axes for different species in Fig. 3b. 15. Line 250. What are your thoughts about SST? Why don't you see how emissions at a particular location has changed with SST in the last 30 years? Add SST to Figure 8

We thank the reviewer for these questions. PMOA emissions are primarily driven by 10-m wind speed, open ocean fraction (1-SIC), SST and OMF (as a proxy for marine productivity). We examined their linear correlation with emission fluxes, and the results are shown in Table 2, Section 3.3.2 of the revised manuscript (see lines 408-415). This table indicates that while 10-m winds and SIC strongly modulate emissions in Arctic sub-regions, SST and OMF have a lesser influence, generally showing a moderate correlation with emissions. The correlation between emissions and SST is slightly lower than that of SIC in all cases.

The SST time series was included in Figure 8. In addition, Table D1 in the supplement summarises the correlation between anomalies and emission drivers. In the manuscript, the discussion of SST time series was included in lines 508-512 and lines 523-526.

16. Why not add years till 2024 to make the data more up-to-date, since it's modelling and does not require observation data availability?

We agree that using the most recent years would bring the dataset to the latest possible state. However, the FESOM-REcoM model output used for this study was only available until 2019; simulations beyond this year are not yet available. Nonetheless, the 30-year period (1990–2019) used in this study corresponds to the standard climatological period commonly applied in climate research (WMO, 2017). Moreover, we believe that extending this period would not significantly change the overall outcome of our study.

**References**

Abdul-Razzak, H. and Ghan, S. J.: A parameterization of aerosol activation: 2. Multiple aerosol types, Journal of Geophysical Research: Atmospheres, 105, 6837–6844, https://doi.org/10.1029/1999JD901161, 2000.

Alpert, P. A., Kilthau, W. P., O'Brien, R. E., Moffet, R. C., Gilles, M. K., Wang, B., Laskin, A., Aller, J. Y., and Knopf, D. A.: Ice-nucleating agents in sea spray aerosol identified and quantified with a holistic multimodal freezing model, Science Advances, 8, https://doi.org/10.1126/sciadv.abq6842, 2022.

- Bigg, E. K. and Leck, C.: Cloud-active particles over the central Arctic Ocean, Journal of Geophysical Research: Atmospheres, 106, 32155–32166, https://doi.org/10.1029/1999JD901152, 2001.
- Cavalli, F., Facchini, M. C., Decesari, S., Mircea, M., Emblico, L., Fuzzi, S., Ceburnis, D., Yoon, Y. J., O'Dowd, C. D., Putaud, J., and Dell'Acqua, A.: Advances in characterization of size-resolved organic matter in marine aerosol over the North Atlantic, Journal of Geophysical Research: Atmospheres, 109, https://doi.org/10.1029/2004JD005137, 2004.
- Creamean, J. M., Barry, K., Hill, T. C. J., Hume, C., DeMott, P. J., Shupe, M. D., Dahlke, S., Willmes, S., Schmale, J., Beck, I., Hoppe, C. J. M., Fong, A., Chamberlain, E., Bowman, J., Scharien, R., and Persson, O.: Annual cycle observations of aerosols capable of ice formation in central Arctic clouds, Nature Communications, 13, 3537, https://doi.org/10.1038/s41467-022-31182-x, 2022.
- Facchini, M. C., Rinaldi, M., Decesari, S., Carbone, C., Finessi, E., Mircea, M., Fuzzi, S., Ceburnis, D., Flanagan, R., Nilsson, E. D., de Leeuw, G., Martino, M., Woeltjen, J., and O'Dowd, C. D.: Primary submicron marine aerosol dominated by insoluble organic colloids and aggregates, Geophysical Research Letters, 35, https://doi.org/10.1029/2008GL034210, 2008.
- Hartmann, M., Gong, X., Kecorius, S., van Pinxteren, M., Vogl, T., Welti, A., Wex, H., Zeppenfeld, S., Herrmann, H., Wiedensohler, A., and Stratmann, F.: Terrestrial or marine-indications towards the origin of ice-nucleating particles during melt season in the European Arctic up to 83.7°N, Atmospheric Chemistry and Physics, 21, 11613–11636, https://doi.org/10.5194/acp-21-11613-2021, 2021.
- Hartmann, S., Schrödner, R., Hassett, B. T., Hartmann, M., van Pinxteren, M., Fomba, K. W., Stratmann, F., Herrmann, H., Pöhlker, M., and Zeppenfeld, S.: Polysaccharides–Important Constituents of Ice-Nucleating Particles of Marine Origin, Environmental Science and Technology, <a href="https://doi.org/10.1021/acs.est.4c08014">https://doi.org/10.1021/acs.est.4c08014</a>, 2025.
- Hawkins, L. N. and Russell, L. M.: Polysaccharides, Proteins, and Phytoplankton Fragments: Four Chemically Distinct Types of Marine Primary Organic Aerosol Classified by Single Particle Spectromicroscopy, Advances in Meteorology, 2010, 1–14, https://doi.org/10.1155/2010/612132, 2010.
- Hu, C., Yue, F., Zhan, H., Leung, K. M., Zhang, R., Gu, W., Liu, H., Chen, A., Cao, Y., Wang, X., and Xie, Z.: Homologous series of n-alkanes and fatty acids in the summer atmosphere from the Bering Sea to the western North Pacific, Atmospheric Research, 285, 106633, https://doi.org/10.1016/j.atmosres.2023.106633, 2023.
- Irish, V. E., Elizondo, P., Chen, J., Chou, C., Charette, J., Lizotte, M., Ladino, L. A., Wilson, T. W., Gosselin, M., Murray, B. J., Polishchuk, E., Abbatt, J. P. D., Miller, L. A., and Bertram, A. K.:

- Ice-nucleating particles in Canadian Arctic sea-surface microlayer and bulk seawater, Atmospheric Chemistry and Physics, 17, 10583–10595, https://doi.org/10.5194/acp-17-10583-2017, 2017.
- Jayarathne, T., Sultana, C. M., Lee, C., Malfatti, F., Cox, J. L., Pendergraft, M. A., Moore, K. A., Azam, F., Tivanski, A. V., Cappa, C. D., Bertram, T. H., Grassian, V. H., Prather, K. A., and Stone, E. A.: Enrichment of Saccharides and Divalent Cations in Sea Spray Aerosol During Two Phytoplankton Blooms, Environmental Science and Technology, 50, 11511–11520, https://doi.org/10.1021/acs.est.6b02988, 2016.
- Jayarathne, T., Gamage, D. K., Prather, K. A., and Stone, E. A.: Enrichment of saccharides at the air–water interface: a quantitative comparison of sea surface microlayer and foam, Environmental Chemistry, 19, 506–516, https://doi.org/10.1071/EN22094, 2023.
- Kahru, M., Brotas, V., Manzano-Sarabia, M., and Mitchell, B. G.: Are phytoplankton blooms occurring earlier in the Arctic?, Global Change Biology, 17, 1733–1739, https://doi.org/10.1111/j. 1365-2486.2010.02312.x, 2011.
- Keene, W. C., Maring, H., Maben, J. R., Kieber, D. J., Pszenny, A. A. P., Dahl, E. E., Izaguirre, M. A., Davis, A. J., Long, M. S., Zhou, X., Smoydzin, L., and Sander, R.: Chemical and physical characteristics of nascent aerosols produced by bursting bubbles at a model air-sea interface, Journal of Geophysical Research: Atmospheres, 112, https://doi.org/10.1029/2007JD008464, 2007.
- Leck, C. and Bigg, E. K.: Source and evolution of the marine aerosol—A new perspective, Geophysical Research Letters, 32, https://doi.org/10.1029/2005GL023651, 2005.
- Leon-Marcos, A., Zeising, M., van Pinxteren, M., Zeppenfeld, S., Bracher, A., Barbaro, E., Engel, A., Feltracco, M., Tegen, I., and Heinold, B.: Modelling emission and transport of key components of primary marine organic aerosol using the global aerosol-climate model ECHAM6.3-HAM2.3, Geoscientific Model Development, 18, 4183-4213, https://doi.org/10.5194/gmd-18-4183-2025, 2025.
- Manizza, M., Carroll, D., Menemenlis, D., Zhang, H., and Miller, C. E.: Modeling the Recent Changes of Phytoplankton Blooms Dynamics in the Arctic Ocean, Journal of Geophysical Research: Oceans, 128, https://doi.org/10.1029/2022JC019152, 2023.
- McCluskey, C. S., Ovadnevaite, J., Rinaldi, M., Atkinson, J., Belosi, F., Ceburnis, D., Marullo, S., Hill, T. C. J., Lohmann, U., Kanji, Z. A., O'Dowd, C., Kreidenweis, S. M., and DeMott, P. J.: Marine and Terrestrial Organic Ice-Nucleating Particles in Pristine Marine to Continentally Influenced Northeast Atlantic Air Masses, Journal of Geophysical Research: Atmospheres, 123, 6196–6212, https://doi.org/10.1029/2017JD028033, 2018.

- Porter, G. C. E., Adams, M. P., Brooks, I. M., Ickes, L., Karlsson, L., Leck, C., Salter, M. E., Schmale,
  J., Siegel, K., Sikora, S. N. F., Tarn, M. D., Vüllers, J., Wernli, H., Zieger, P., Zinke, J., and Murray,
  B. J.: Highly Active Ice-Nucleating Particles at the Summer North Pole, Journal of Geophysical Research: Atmospheres, 127, https://doi.org/10.1029/2021JD036059, 2022.
- Rastelli, E., Corinaldesi, C., Dell'Anno, A., Martire, M. L., Greco, S., Facchini, M. C., Rinaldi, M., O'Dowd, C., Ceburnis, D., and Danovaro, R.: Transfer of labile organic matter and microbes from the ocean surface to the marine aerosol: an experimental approach, Scientific Reports, 7, 11475, https://doi.org/10.1038/s41598-017-10563-z, 2017.
- Russell, L. M., Hawkins, L. N., Frossard, A. A., Quinn, P. K., and Bates, T. S.: Carbohydrate-like composition of submicron atmospheric particles and their production from ocean bubble bursting, Proceedings of the National Academy of Sciences, 107, 6652–6657, https://doi.org/10.1073/pnas.0908905107, 2010.
- Tegen, I., Neubauer, D., Ferrachat, S., Drian, C. S.-L., Bey, I., Schutgens, N., Stier, P., Watson-Parris,
  D., Stanelle, T., Schmidt, H., Rast, S., Kokkola, H., Schultz, M., Schroeder, S., Daskalakis, N.,
  Barthel, S., Heinold, B., and Lohmann, U.: The global aerosol-climate model ECHAM6.3-HAM2.3
  Part 1: Aerosol evaluation, Geoscientific Model Development, 12, 1643-1677, https://doi.org/10.5194/gmd-12-1643-2019, 2019.
- Triesch, N., Pinxteren, M. V., Engel, A., and Herrmann, H.: Concerted measurements of free amino acids at the Cabo Verde islands: High enrichments in submicron sea spray aerosol particles and cloud droplets, Atmospheric Chemistry and Physics, 21, 163–181, https://doi.org/10.5194/acp-21-163-2021, 2021a.
- Triesch, N., Pinxteren, M. V., Frka, S., Stolle, C., Spranger, T., Hoffmann, E. H., Gong, X., Wex, H., Schulz-Bull, D., Gasparovic, B., and Herrmann, H.: Concerted measurements of lipids in seawater and on submicrometer aerosol particles at the Cabo Verde islands: Biogenic sources, selective transfer and high enrichments, Atmospheric Chemistry and Physics, 21, 4267–4283, https://doi.org/10.5194/acp-21-4267-2021, 2021b.
- van Pinxteren, M., Zeppenfeld, S., Fomba, K. W., Triesch, N., Frka, S., and Herrmann, H.: Amino acids, carbohydrates, and lipids in the tropical oligotrophic Atlantic Ocean: sea-to-air transfer and atmospheric in situ formation, Atmospheric Chemistry and Physics, 23, 6571–6590, https://doi.org/10.5194/acp-23-6571-2023, 2023.
- WMO: Weather Meteorological Organization, WMO Guidelines on the Calculation of Climate Normals, Tech. rep., https://community.wmo.int/en/wmo-climatological-normals, 2017.

- Zeising, M., Oziel, L., Thoms, S., Özgür Gürses, Hauck, J., Heinold, B., Losa, S. N., van Pinxteren, M., Völker, C., Zeppenfeld, S., and Bracher, A.: Assessment of transparent exopolymer particles in the Arctic Ocean implemented into the coupled ocean—sea ice—biogeochemistry model FESOM2.1–REcoM3, https://doi.org/10.5194/egusphere-2025-4190, 2025.
- Zeppenfeld, S., Pinxteren, M. V., Pinxteren, D. V., Wex, H., Berdalet, E., Vaqué, D., Dall'osto, M., and Herrmann, H.: Aerosol Marine Primary Carbohydrates and Atmospheric Transformation in the Western Antarctic Peninsula, ACS Earth and Space Chemistry, 5, 1032–1047, https://doi.org/10.1021/acsearthspacechem.0c00351, 2021.
- Zhao, H., Matsuoka, A., Manizza, M., and Winter, A.: Recent Changes of Phytoplankton Bloom Phenology in the Northern High-Latitude Oceans (2003–2020), Journal of Geophysical Research: Oceans, 127, https://doi.org/10.1029/2021JC018346, 2022.

---

## Author Comment (AC3)

**Response to RC1: 'Comment on egusphere-2025-2829'**

**October 22, 2025**

We are grateful to the reviewer for taking the time to review our manuscript and for offering such valuable feedback. Your observations and helpful recommendations contributed to enhancing the clarity and quality of our work. Please find below the response (in blue) to your comments (in black)

The authors have undergone an extensive modelling study to understand the impact of several factors (sea ice extent, wind speed, SST) on the emission of speciated PMOA over the last several decades. They have done the analysis on a regional basis and searched for trends relating decreases in sea ice area to increases in PMOA emissions. As they discuss in section 5, all of this is very difficult to do considering the lack of PMOA seawater and atmospheric aerosol data, especially speciated, in the Arctic. Given the lack of data and the many uncertainties involved, the paper advances the understanding of factors controlling the emission, seasonality, and trends in Arctic PMOA.

- 1. Lines 184 185: Were open leads and melt ponds considered in estimating PMOA concentrations? Leads and melt ponds were not included in the model; we have added a sentence at line 186 to explicitly state this: "However, in the present study, open leads and melt ponds are not included in the model simulations."
- 2. Table 1: Is an analysis of annually averaged data justified since there is so much seasonality in the factors controlling emission flux, burden, and deposition?

We agree, annually averaged data would not be justified in this case given the strong seasonality shown, for example, in Fig. 5. However, Table 1 does not show the annual average values, but rather the multi-year average over a period of 15 or 30 years of total annual emissions over the Arctic. Since the table header was misleading, we changed it as follows: "Table 1: Total emission flux, atmospheric burden, and deposition of marine aerosol particles, calculated by summing daily values across all Arctic grid cells and averaging yearly totals over the two 15-year periods and the full 30-year period." Following this comment, and for consistency with Table 1 and Table

- 3, we also decided to show total seasonal emissions as just explained in Fig. 5 and Fig. 11 for July-August-September, and Fig. F5 for April-May-June. Additionally, for better alignment with Fig. 9, Fig. 10 shows the trends of regionally averaged emission fluxes.
- 3. Lines 318: Is there a reference for the SST correction factor for the SS emission flux?

  The sea surface temperature (SST) dependence of the sea spray emission flux is implemented following Sofiev et al. (2011). This reference has now been included in the text.
- 4. Figure 5: The order of figures for e) and f) is different than what is stated in the caption.

  We have corrected this error in the figure caption.
- 5. Lines 339 341: It is stated on lines 301 302 that SS emissions from the model are a sum of both the accumulation and coarse modes. Earlier in the paper it says that PMOA fluxes are based on SS fluxes which are based on temperature. Yet, it is stated here (lines 339 341) that the SST correction factor used in SS model simulations remains relatively similar for the accumulation mode. Is only the SST correction for the accumulation mode used for the speciated PMOA fluxes? It seems confusing if the speciated PMOA fluxes are coming from size independent (accumulation + coarse modes) SST emissions but size a dependent (accumulation mode only) SST correction.

PMOA is only emitted into the accumulation mode, based on SS emission of the same mode, which depends on SST. PMOA is not emitted into the coarse mode directly, and the mass emission fluxes of PMOA do not encompass the coarse-size particles. In contrast, SS is additionally emitted into the coarse mode. Hence, the SST correction only affects the accumulation-mode PMOA emissions while for SS, it influences both the finer and coarse modes.

In the model output, the emission fluxes are not mode-dependent. Therefore, in lines 301 - 302, we specify that the SS emissions are not accumulation mode alone but also coarse mode.

To make these aspects clearer, we modified lines 339-341 as follows: "Overall, SST ranges between -2 to 6 °C. Within this temperature range, the Sofiev et al. (2011) SST correction factor used in the SS model representation remains relatively similar for the particles in the accumulation mode, which is the only size class contributing to PMOA emissions. Therefore, in this case, SST has a lesser effect on marine emissions."

- 6. Figure 8: Should this be Beaufort Sea for (b)?

  That is correct, it should be Beaufort Sea. We have changed this in the figure caption.
- Line 611: Change "along" to "alone".
   Thanks for pointing this out. We corrected it.

**References**

Sofiev, M., Soares, J., Prank, M., de Leeuw, G., and Kukkonen, J.: A regional-to-global model of emission and transport of sea salt particles in the atmosphere, Journal of Geophysical Research: Atmospheres, 116, https://doi.org/10.1029/2010JD014713, 2011.

---

## Referee Report (RR1)

This work uses atmospheric modelling combined with a marine biogeochemical model to investigate the drivers and evolution of primary marine organic aerosols in the Arctic, in the context of the changing sea-ice. The topic is very relevant and an important contribution to the community. The methods are well described and the figures are overall clear.

My main concern is the very high level of detail provided in the manuscript, which sometimes makes sentences and paragraphs hard to follow, and distracts from the important messages of the paper. I provide a few examples below, along with a few minor/specific comments. The scientific contents of the paper are otherwise good and it deserves publication in ACP.

Specific comments

- L177: "The biomolecule ocean concentration serves as boundary condition for ECHAM-HAM, as explained in the previous section" - if it is explained in the previous section, no need to say it again. Please try to get rid of all similar occurrences of "as previously explained" throughout the manuscript.

- L186–195: please try to find a more direct way to explain your sea-ice mask. I am not sure I understand it. Suggestion of rewording "[… prevents bubble bursting at the surface.] Although sea spray emissions can occur in the marginal ice zone and within the sea ice pack through open leads and melt ponds (REFS), these sources are not considered in this study for lack of model of their emission fluxes. Therefore, because we cannot include these sources, we apply a sea ice mask that considers only the open ocean grid cells (SIC<10%, Arrigo et al., 2008) when, and only when, we present average parameters over the Arctic. [Additionally, for a more profound understanding…]" This is 78 words instead of 162 and I think this conveys the same message but in a more direct and clearer way. I encourage the authors to try to do the same exercise for the excerpts referenced hereafter and more generally throughout the manuscript. The quality and impact of the manuscript would be greatly improved.

- L210: "… for the compounds simulated in the present study…" is unnecessary, it is clearly understood that this is going to be the compounds simulated in the study. Instead simply say "The simulated biomolecule ocean concentrations are shown in Fig. 2 as a multi-annual…" is enough and less distracting. Same as above, try to get rid of unnecessary pieces of sentences.

- Section 3.2 - Although this part is interesting, an ACP paper should focus on the atmosphere. I think this section should therefore be trimmed to the very minimum, with as few numbers as possible and only the main important results that will shed light on the analysis of the atmosphere that is coming afterwards. Also you talk a lot about the sea ice and central Arctic in this section but since emissions into the atmosphere are only considered for open ocean this part is not very relevant for the analysis. This section could largely go as a supplement.

L342—357: this description of SIC and SST is too detailed and does not bring much to the paper. Please condense/synthesise to keep only the main information relevant

for the next part on aerosols.

L358—371: this is about sea salt emissions, not specifically organics, but in the next paragraph you say that organics and SS have different seasonalities. Therefore I wonder if this paragraph and Table 2 should be made organics specific and not sea-salt oriented only. In addition, since the relationship between wind speed and SS emissions is not linear, I do not expect a Pearson correlation coefficient to accurately represent the influence of wind speed on SS. I would use Spearman correlation instead. How is this correlation computed anyway? Is it on the 12-hourly output values of the model?

L422–424: I do not understand what you are trying to say in these two sentences

Section 4.1: again this part focuses a lot on oceanic concentrations of precursors, which I agree the paper should address, but not to that extent for an ACP publication. I would expect an analysis where oceanic biological activity is considered a driver of atmospheric emissions and is therefore described not as the main object of study but more as an explanatory variable for emissions. For example, schematically I would expect: "We observe a trend in emissions of organics in the XX sea, which is driven by changes in biological activity in the ocean, related to changes in SIC…". Therefore I would summarise 4.1 down to essential information, maybe not with detailed regional analysis, to offer context for the analysis of emissions that follows, but not much more.

L523—525: how do you compute average SST and how is it affected by changes in sea-ice? You say there is a positive correlation between SST and emission anomalies but SST is known to have a relatively small (and still debated whether it is positive or negative at low temperature) effect on emissions. Isn't this correlation you find simply because SST trend is related to SIC trend?

L568: I do not understand this sentence.

Comments on figures and tables

- Table 1: The deposition flux of organics is systematically larger than the emission flux. For sea salt it is the opposite. Can you explain / comment on this difference? SS and organics are co-emitted so does that mean that transport/activation in clouds is different? How are PCHO, DCAA and PL activated as CCN/INP? Same as SS?

- Table D1 contains the same information as Table 2.

- Figure 7: I do not understand what dot size corresponds to.

- Table 3: I assume this is **surface** concentration? This should be clearly stated.

---

## Author Response (AR2)

**Response to Report 2 and Report 3**

December 6, 2025

**Report 2**

We are grateful to the reviewer for revising the resubmitted version of our manuscript and for offering further feedback. Your observations helped enhance the clarity and quality of our work. Please find below the response (in blue) to your comments (in black)

Few minor revisions:

1. line 10, "Total aerosol emission and burden over the Arctic increased by at least 12% and 4%, respectively, between 1990–2004 and 2005–2019." May be write respectively in the end if you mean to say 12% is till 2004 and 4% is from 2005

    We thank the reviewer for providing this feedback, as we realised the sentence must be revised for clarity. Note that we refer to the increase in total annual emissions and burden from the first half (averaged over 1990-2004) to the second half (averaged over 2005-2019) of the simulated period. In other words, the PMOA total emissions increased by 12%, and the burden increased by 4%, from the period 1990-2004 to 2005-2019. To clarify this further, we revised the sentence as follows: " Total PMOA emissions increased by about 12%, and the burden rose by 4% between 1990–2004 and 2005–2019."

2. Abstract: "Positive emission anomalies have become more frequent over the past 15 years, indicating an overall upward trend.". mention positive emission anomalies

of what? Its confusing because you also say how aerosol emissions are at 12%
increase and at 4% increase later. Also why is 12% and 4% thing important if
your focus is on PMOA?

Here we refer to the PMOA positive anomalies calculated with respect to the
multiannual mean. We added PMOA to the sentence "Positive PMOA emission
anomalies . . . ", consequently. As explained in the previous response, 12 % refers to
the increase of PMOA total marine emissions from 1990-2004 to 2005-2019, while
4% corresponds to the change in PMOA burden. We believe these percentages
convey valuable information about the changes in PMOA emissions and burden
based on the 15-year mean, with a significant increase in aerosol quantities observed
through 2005-2019 with respect to the earlier period 1990-2004.

3. Abstract: "However, changes vary across biomolecular types and Arctic subre-
gions, with PCHO showing the largest relative increase, with 1.13% and 0.8% per
year for the emissions and aerosol concentration, respectively." I think you should
also mention the trends for amino acids and polar lipids too in the abstract.

We understand that, including the trends for amino acids and polar lipids in the
abstract, provides a more complete summary of our results. However, given the
rigorous limit of words to include in the Abstracts of ACP papers (250 words), we
are restrained from including further details in the Abstract. Hence, we decided
to present the most relevant results for the total PMOA and highlight species-
specific differences by emphasising that the most pronounced changes are observed
for PCHO.

4. Section 3.1 "Biomolecule ocean concentration is shown in Fig. 2 for the compounds
simulated in the present study as multiannual average over the period 1990–2019".
You mentioned in figure caption that's its from May-Sep every year but it would
be nice if you mention the months in the text too.

Thanks for the recommendation, we added this in line 212 ". . . average from May
to September over the period . . . "

5. If $DCAA_{sw}$ is a fraction of $PCHO_{sw}$ then why the need for $DCAA_{sw}$ separately? Cant $PCHO_{sw}$ represent it?

In seawater, $PCHO_{sw}$ distribution can represent that of $DCAA_{sw}$; however, in terms of carbon concentration, considering $DCAA_{sw}$ in addition to $PCHO_{sw}$ is necessary. This is especially relevant for aerosols, since protein-like and polysaccharide-like compounds have different physicochemical characteristics that regulate the aerosolisation. The former has a higher surface affinity than the latter (van Pinxteren et al., 2023) and is more readily transferred to aerosols during bubble bursting. In our model configuration, these properties are considered in the emission scheme OCEANFILMS (Burrows et al., 2014) and summarised in Table 1 of Leon-Marcos et al. (2025).

6. Line 218: "These differences also vary along throughout the year" show, may be in supplementary?

We strongly agree with the reviewer, as the seasonality, which also shows the differences throughout the year, is presented in the following section. Hence, we excluded this sentence from the manuscript.

**Report 3**

We are grateful to the reviewer for taking the time to review our manuscript and for offering such valuable feedback. Your observations and helpful recommendations contributed to enhancing the clarity and quality of our work. Please find below the response (in blue) to your comments (in black)

This work uses atmospheric modelling combined with a marine biogeochemical model to investigate the drivers and evolution of primary marine organic aerosols in the Arctic, in the context of the changing sea-ice. The topic is very relevant and an important contribution to the community. The methods are well described and the figures are overall clear.

My main concern is the very high level of detail provided in the manuscript, which sometimes makes sentences and paragraphs hard to follow, and distracts from the important messages of the paper. I provide a few examples below, along with a few minor/specific comments. The scientific contents of the paper are otherwise good and it deserves publication in ACP.

Specific comments:

1. L177: "The biomolecule ocean concentration serves as boundary condition for ECHAM-HAM, as explained in the previous section" - if it is explained in the previous section, no need to say it again. Please try to get rid of all similar occurrences of "as previously explained" throughout the manuscript.

   We thank the reviewer for highlighting this. We revised the manuscript and removed these occurrences.

2. L186–195: please try to find a more direct way to explain your sea-ice mask. I am

not sure I understand it. Suggestion of rewording "[... prevents bubble bursting at the surface.] Although sea spray emissions can occur in the marginal ice zone and within the sea ice pack through open leads and melt ponds (REFS), these sources are not considered in this study for lack of model of their emission fluxes. Therefore, because we cannot include these sources, we apply a sea ice mask that considers only the open ocean grid cells (SIC<10%, Arrigo et al., 2008) when, and only when, we present average parameters over the Arctic. [Additionally, for a more profound understanding...]" This is 78 words instead of 162 and I think this conveys the same message but in a more direct and clearer way. I encourage the authors to try to do the same exercise for the excerpts referenced hereafter and more generally throughout the manuscript. The quality and impact of the manuscript would be greatly improved.

We greatly appreciate this excellent suggestion that significantly improved the clarity of the explanation of the sea-ice mask. Consequently, we changed the text as follows: "...prevents bubble bursting at the surface. Although sea spray emissions can occur in the marginal ice zone and within the Arctic sea ice pack from open leads and melt ponds (Leck and Bigg, 2005; Willmes and Heinemann, 2015; Zhang et al., 2018; Rolph et al., 2020), these sources are not considered in this study. Consequently, since these sources cannot be represented, we apply a sea-ice mask that restricts the analysis to open-ocean grid cells (SIC<10 %; Arrigo et al., 2008) exclusively for calculations of average marine parameters and aerosol OMF over the Arctic. Note that the mask does not apply to the use of the biomolecule ocean concentrations as bottom boundary condition within the ECHAM-HAM simulations. Additionally ..."

3. L210: "... for the compounds simulated in the present study..." is unnecessary, it is clearly understood that this is going to be the compounds simulated in the study. Instead simply say "The simulated biomolecule ocean concentrations are shown in Fig. 2 as a multi-annual..." is enough and less distracting. Same as

above, try to get rid of unnecessary pieces of sentences.

Thank you for this suggestion. We modified this sentence as suggested and removed unnecessary parts similar to this one.

4. Section 3.2 - Although this part is interesting, an ACP paper should focus on the atmosphere. I think this section should therefore be trimmed to the very minimum, with as few numbers as possible and only the main important results that will shed light on the analysis of the atmosphere that is coming afterwards. Also you talk a lot about the sea ice and central Arctic in this section but since emissions into the atmosphere are only considered for open ocean this part is not very relevant for the analysis. This section could largely go as a supplement.

We understand the reviewer's concern regarding the extent of the sections related to marine organic aerosol precursors in seawater. This section (Section 3.3.2) lays the foundation for a deeper understanding of the occurrence of organic marine emissions and especially their seasonality. The analysis of marine biological productivity as an emission driver is essential, as it provides a differentiated picture of Arctic subregions that has not been studied to this degree. Because these processes occur at the ocean–atmosphere interface, they cannot be understood solely by considering the atmosphere. Nonetheless, following the reviewer recommendation, we trimmed this section considerably, shifting the regional seasonality from Figure 3(c, d) to the supplement (Fig. S2). The text was also condensed to some extent, yet it still highlights regional differences within the Arctic that are essential to understand PMOA occurrence throughout the manuscript better.

5. L342—357: this description of SIC and SST is too detailed and does not bring much to the paper. Please condense/synthesise to keep only the main information relevant for the next part on aerosols.

We have condensed this description and especially provided fewer details on the regional changes of SIC.

6. L358—371: this is about sea salt emissions, not specifically organics, but in the next paragraph you say that organics and SS have different seasonalities. Therefore I wonder if this paragraph and Table 2 should be made organics specific and not sea-salt oriented only. In addition, since the relationship between wind speed and SS emissions is not linear, I do not expect a Pearson correlation coefficient to accurately represent the influence of wind speed on SS. I would use Spearman correlation instead. How is this correlation computed anyway? Is it on the 12-hourly output values of the model?

By discussing the seasonality of SS in this paragraph (L358—371) and later the seasonality of PMOA species, we intend to show that although PMOA is co-emitted with SS, the discrepancies in the seasonality show how the marine biological productivity regulates the PMOA emission seasonal cycle. However, we agree that the SS emission description is lengthy and could be shortened by presenting it more in contrast to the PMOA emissions. We did this and removed unnecessary text from L358—371. Table 2 is now organics-specific only. We specified this in the table header: "Spearman correlation coefficients between total PMOA emission flux and emission drivers...". Regarding the correlation coefficient, we agree that the Spearman correlation would better reflect the relationship between emissions and drivers, given their non-linear relationship (e.g. 10-m wind speed). Given your recommendation, we computed the Spearman correlation and updated the values in Table 2. We had used temporally and spatially averaged emissions and drivers for each year. However, because mean values do not capture the variability, we used daily emission and driver values to compute Spearman correlations. The updated Table 2 summarises the results per season. Lines 365-376 of the revised manuscript discuss these results. .

7. L422–424: I do not understand what you are trying to say in these two sentences

To focus our analysis on regions potentially ice-free where marine emissions could

occur, the maps of the trends of biomolecule ocean concentration, the minimum SIC for the season, is represented. The SIC overlaps the trends of marine biomolecules, visually excluding areas potentially permanently covered by ice. Hence, we rewrote these lines as follows: "... FESOM-REcoM are also included. To restrict our analysis to potentially ice-free regions where marine emissions may occur, we overlaid the seasonal minimum SIC on trends of ocean organic quantities, thereby visually excluding areas that are likely permanently ice-covered. In addition, ..."

8. Section 4.1: again this part focuses a lot on oceanic concentrations of precursors, which I agree the paper should address, but not to that extent for an ACP publication. I would expect an analysis where oceanic biological activity is considered a driver of atmospheric emissions and is therefore described not as the main object of study but more as an explanatory variable for emissions. For example, schematically I would expect: "We observe a trend in emissions of organics in the XX sea, which is driven by changes in biological activity in the ocean, related to changes in SIC...". Therefore I would summarise 4.1 down to essential information, maybe not with detailed regional analysis, to offer context for the analysis of emissions that follows, but not much more.

Following the reviewer's recommendation, we decided to remove Figs. 7 and S4, which show the regional analysis of all biomolecules across Arctic subregions, along with the related text. With this, the section is significantly reduced, still briefly mentioning the most significant regional differences relevant for marine organic emissions.

9. L523—525: how do you compute average SST and how is it affected by changes in sea-ice? You say there is a positive correlation between SST and emission anomalies but SST is known to have a relatively small (and still debated whether it is positive or negative at low temperature) effect on emissions. Isn't this correlation you find simply because SST trend is related to SIC trend?

Average SST for a specific region is computed as a weighted mean based on the grid cell area. Monthly mean values of SST and SIC are prescribed as boundary conditions in the ECHAM-HAM model, using data from the Atmospheric Model Intercomparison Project (AMIP) (Taylor et al., 2000). Because they are derived from monthly mean observations, SST and SIC provide information on actual conditions. Hence, as thinning sea ice and sea ice loss contribute to a positive ice albedo feedback, SST will consequently increase. We agree that SST have a relatively small direct influence on emissions, making it even more challenging to identify at low temperatures and for small particle sizes (e.g. Barthel et al., 2019). However, note that rising ocean temperatures partly drive higher marine biological production (Wu et al., 2025), as the SST–productivity relationship is more nuanced than the SIC–productivity. Therefore, modifying biomolecule abundance and then PMOA emissions. Note that increasing SST is altering this relationship, strengthening upper-ocean stratification and reducing vertical nutrient flux (Wu et al., 2025; Noh et al., 2024). To assess the correlation between marine emissions and ocean drivers, we computed the Spearman correlations in the Arctic during summer and spring (see Fig.1 further below). In areas seasonally covered by sea ice, the positive correlation between emissions and SST is probably influenced by the natural dependence between SIC and SST (as evidenced by the similar distributions in the correlation maps of SIC and SST). In this case, higher SST enhances vertical mixing in newly ice-free shelf and marginal ice zone regions, increasing nutrient supply and biomolecule production, indirectly contributing to the strong positive correlation with OMF, especially in spring. In areas with very low or absent sea ice (e.g., the Barents Sea, Norwegian, and eastern Greenland seas), the correlation between SST and emissions is less affected by SIC, with positive and negative correlations in spring and summer, respectively.

In the manuscript, we emphasised that emissions are only moderately correlated to SST (see line 464 of the revised manuscript) and deleted line 523. Moreover, we added in line 329 (section 3.3.2): "Nonetheless, ocean temperatures modulate hydrographic conditions, strongly affecting marine productivity and, in turn, PMOA
emissions"

10. L568: I do not understand this sentence.

We realised this sentence was difficult to understand and rewrote it as follows:
Overall, the spatial distributions of marine organic species across the Arctic are in
close agreement.

Comments on figures and tables

1. Table 1: The deposition flux of organics is systematically larger than the emission
flux. For sea salt it is the opposite. Can you explain / comment on this difference?
SS and organics are co-emitted so does that mean that transport/activation in
clouds is different? How are PCHO, DCAA and PL activated as CCN/INP? Same
as SS?

That is an excellent observation. This is related to how the species are treated in
the model. While PMOA is co-emitted with SS in the accumulation mode only,
SS is also emitted in the coarse mode. However, after emission, PMOA could
grow into the coarse mode through coagulation or condensation. Hence, the total
aerosol mass available for activation as CCN and deposition is larger than that
emitted. This explains why the deposition flux of organics is systematically larger
than the emission flux. In contrast, because sea salt mass is strictly defined in the
accumulation and coarse modes since emission, the deposition flux could be lower
than the emissions. As for the transport and activation in clouds, inorganic and
organic species within the same mode are identical. In the ECHAM6.3-HAM2.3
model, CCN is calculated for each aerosol mode following Abdul-Razzak and Ghan
(2000) and encompasses the contribution of all species within a mode. However,
the ice nucleation potential of SS or PMOA has not yet been implemented in the
model.

2. Table D1 contains the same information as Table 2.

Thank you for pointing this out. Nonetheless, this table was removed since the values are very similar to those in Table 2 and do not add new information.

3. Figure 7: I do not understand what dot size corresponds to.

The size indicates a larger grid fraction in %, and it is in accordance with the colour bar. The larger the circle, the higher the percentage. Although not as visually obvious in some cases, yellow circles indicate higher percentages than the green or purple ones.

4. Table 3: I assume this is surface concentration? This should be clearly stated.

Yes, the aerosol concentration values correspond to the lowermost model layer. We have clearly stated it in the Table header ("...mass flux and near-surface average PMOA concentration...") and in line 528.

**Figures**

[Figure]

Figure 1: Spearman correlation of total PMOA emission and emission drivers over April-May-June (AMJ) and July-August-September (JAS) during the simulated period 1990–2019. Only statistically significant values are shown (t-test, p-value<0.05).

**References**

Abdul-Razzak, H. and Ghan, S. J.: A parameterization of aerosol activation: 2. Multiple aerosol types, Journal of Geophysical Research: Atmospheres, 105, 6837–6844, https://doi.org/10.1029/1999JD901161, 2000.

Arrigo, K. R., van Dijken, G., and Pabi, S.: Impact of a shrinking Arctic ice cover on marine primary production, Geophysical Research Letters, 35, https://doi.org/10.1029/2008GL035028, 2008.

Barthel, S., Tegen, I., and Wolke, R.: Do new sea spray aerosol source functions improve the results of a regional aerosol model?, Atmospheric Environment, 198, 265–278, https://doi.org/10.1016/j.atmosenv.2018.10.016, 2019.

Burrows, S. M., Ogunro, O., Frossard, A. A., Russell, L. M., Rasch, P. J., and Elliott, S. M.: A physically based framework for modeling the organic fractionation of sea spray aerosol from bubble film Langmuir equilibria, Atmospheric Chemistry and Physics, 14, 13 601–13 629, https://doi.org/10.5194/acp-14-13601-2014, 2014.

Leck, C. and Bigg, E. K.: Biogenic particles in the surface microlayer and overlaying atmosphere in the central Arctic Ocean during summer, Tellus B: Chemical and Physical Meteorology, 57, 305, https://doi.org/10.3402/tellusb.v57i4.16546, 2005.

Leon-Marcos, A., Zeising, M., van Pinxteren, M., Zeppenfeld, S., Bracher, A., Barbaro, E., Engel, A., Feltracco, M., Tegen, I., and Heinold, B.: Modelling emission and transport of key components of primary marine organic aerosol using the global aerosol–climate model ECHAM6.3–HAM2.3, Geoscientific Model Development, 18, 4183–4213, https://doi.org/10.5194/gmd-18-4183-2025, 2025.

Noh, K., Oh, J., Lim, H., Song, H., and Kug, J.: Role of Atlantification in Enhanced Primary Productivity in the Barents Sea, Earth's Future, 12, https://doi.org/10.1029/2023EF003709, 2024.

Rolph, R. J., Feltham, D. L., and Schröder, D.: Changes of the Arctic marginal ice

zone during the satellite era, The Cryosphere, 14, 1971–1984, https://doi.org/10.5194/tc-14-1971-2020, 2020.

Taylor, K. E., Williamson, D. L., and Zwiers, F. W.: The Sea Surface Temperature and Sea-Ice Concentration Boundary Conditions for AMIP II Simulations, Program for Climate Model Diagnosis and Intercomparison (PCMDI) Report 60, Lawrence Livermore National Laboratory, Livermore, California, 2000.

van Pinxteren, M., Zeppenfeld, S., Fomba, K. W., Triesch, N., Frka, S., and Herrmann, H.: Amino acids, carbohydrates, and lipids in the tropical oligotrophic Atlantic Ocean: sea-to-air transfer and atmospheric in situ formation, Atmospheric Chemistry and Physics, 23, 6571–6590, https://doi.org/10.5194/acp-23-6571-2023, 2023.

Willmes, S. and Heinemann, G.: Sea-Ice Wintertime Lead Frequencies and Regional Characteristics in the Arctic, 2003–2015, Remote Sensing, 8, 4, https://doi.org/10.3390/rs8010004, 2015.

Wu, M., Hu, Y., Le, C., Lin, F., Gu, T., Deng, P., Jiang, Z.-P., Wang, K., Chen, Y., Wang, W.-L., Lan, M., Lei, R., He, J., Qi, D., Ouyang, Z., Lehrter, J. C., Hu, C., and Cai, W.-J.: Sea Ice Loss leads to regime shifts in the arctic biological pump, Nature Communications, 16, 10 331, https://doi.org/10.1038/s41467-025-65285-y, 2025.

Zhang, J., Schweiger, A., Webster, M., Light, B., Steele, M., Ashjian, C., Campbell, R., and Spitz, Y.: Melt Pond Conditions on Declining Arctic Sea Ice Over 1979–2016: Model Development, Validation, and Results, Journal of Geophysical Research: Oceans, 123, 7983–8003, https://doi.org/10.1029/2018JC014298, 2018.